# Heavy Tails in SGD and Compressibility of Overparametrized Neural Networks

**Melih Barsbey**[*]
Boğaziçi University
melih.barsbey@boun.edu.tr

**Milad Sefidgaran**[*]
LTCI, Télécom Paris, Institut Polytechnique de Paris
milad.sefidgaran@telecom-paris.fr

**Murat A. Erdogdu**
University of Toronto & Vector Institute
erdogdu@cs.toronto.edu

**Gaël Richard**
LTCI, Télécom Paris, Institut Polytechnique de Paris
gael.richard@telecom-paris.fr

**Umut Şimşekli**
INRIA & ENS – PSL Research University
umut.simsekli@inria.fr

## Abstract

Neural network compression techniques have become increasingly popular as they can drastically reduce the storage and computation requirements for very large networks. Recent empirical studies have illustrated that even simple pruning strategies can be surprisingly effective, and several theoretical studies have shown that compressible networks (in specific senses) should achieve a low generalization error. Yet, a theoretical characterization of the underlying causes that make the networks amenable to such simple compression schemes is still missing. In this study, focusing our attention on stochastic gradient descent (SGD), our main contribution is to link compressibility to two recently established properties of SGD: (i) as the network size goes to infinity, the system can converge to a *mean-field* limit, where the network weights behave independently [DBDFŞ20], (ii) for a large step-size/batch-size ratio, the SGD iterates can converge to a *heavy-tailed* stationary distribution [HM20, GŞZ21]. Assuming that both of these phenomena occur *simultaneously*, we prove that the networks are guaranteed to be '$\ell_p$-compressible', and the compression errors of different pruning techniques (magnitude, singular value, or node pruning) become arbitrarily small as the network size increases. We further prove generalization bounds adapted to our theoretical framework, which are consistent with the observation that the generalization error will be lower for more compressible networks. Our theory and numerical study on various neural networks show that large step-size/batch-size ratios introduce heavy tails, which, in combination with overparametrization, result in compressibility.

## 1 Introduction

With the increasing model sizes in deep learning and with its increasing use in low-resource environments, network compression is becoming ever more important. Among many network compression techniques, network pruning has been arguably the most commonly used method [O'N20], and it is rising in popularity and success [BOFG20]. Though various pruning methods are successfully used in practice and their theoretical implications in terms of generalization are increasingly apparent [AGNZ18], a thorough understanding of *why* and *when* neural networks are compressible is lacking.

---

[*]Equal contribution.

35th Conference on Neural Information Processing Systems (NeurIPS 2021).

A common conclusion in pruning research is that overparametrized networks can be greatly compressed by pruning with little to no cost at generalization, including with simple schemes such as magnitude pruning [BOFG20, O'N20]. For example, research on *iterative magnitude pruning* [FC19] demonstrated the possibility of compressing trained deep learning models by iteratively eliciting a much sparser substructure. While it is known that the choice of training hyperparameters such as learning rate affects the performance of such pruning strategies [FDRC20, HJRY20, RFC20], usually such observations are low in granularity and almost never theoretically motivated. Overall, the field lacks a framework to understand why or when a pruning method should be useful [O'N20].

Another strand of research that highlights the importance of understanding network compressibility includes various studies [AGNZ18, SAM+20, SAN20, HJTW21, KLG+21] that presented generalization bounds and/or empirical evidence that imply that the more compressible a network is, the more likely it is to generalize well. The aforementioned bounds are particularly interesting since classical generalization bounds increase with the dimension and hence become irrelevant in high dimensional deep learning settings, and fall short of explaining the generalization behavior of overparametrized neural networks. These results again illustrate the importance of understanding the conditions that give rise to compressibility given their implications regarding generalization.

In this paper, we develop a theoretical framework to address (i) *why and when modern neural networks can be amenable to very simple pruning strategies* and (ii) *how this relates to generalization*. Our theoretical results are based on two recent disparate discoveries regarding deep neural networks trained with the stochastic gradient descent (SGD) algorithm. The first one is the emergence of *heavy-tailed* stationary distributions, which appear when the networks are trained with large learning rates and/or small batch-sizes [HM20, GŞZ21]. The second one is the *propagation of chaos* phenomenon, which indicates that, as the network size goes to infinity, the network weights behave *independently* [MMN18, SS20, DBDFŞ20].

We show that, assuming both of the aforementioned phenomena occur *simultaneously*, fully connected neural networks will be provably compressible in a precise sense, and the compression errors of (i) unstructured global or layer-wise magnitude pruning, (ii) pruning based on singular values of the weight matrices, (iii) and node pruning can be made arbitrarily small for any compression ratio as the dimension increases. Our formulation of network compressibility in terms of '$\ell_p$-compressibility' enables us to access results from *compressed sensing* literature [GCD12, AUM11] to be used in neural network analysis. Moreover, we prove generalization bounds adapted to our framework that agree with existing compression-based generalization bounds [AGNZ18, SAM+20, SAN20] and confirm that compressibility implies better generalization. We conduct experiments on fully connected and convolutional networks and show that the results are in strong accordance with our theory.

Our study reveals an interesting phenomenon: depending on the algorithm hyperparameters, such as learning rate and batch-size, the resulting neural networks might possess different compressibility properties. Under the decoupling effect of propagation of chaos that emerges with overparametrization [DBDFŞ20], the networks become compressible in a way that they are amenable to simple pruning strategies as the tails get heavier, which is shown to depend on the step-size/batch-size ratio [GŞZ21]. Finally, when compressible, the networks become likely to generalize better. In this sense, our results also provide an alternative perspective to the recent theoretical studies that suggest that heavy tails can be beneficial in terms of generalization [ŞSDE20, ZFM+20].

## 2 Preliminaries and Technical Background

**Notation.** Matrices and vectors are denoted by upper- and lower-case bold letters, respectively, *e.g.* $\mathbf{X}$ and $\mathbf{x}$. A sequence of $n$ scalars $x_1, \ldots, x_n$ is shown as $\{x_i\}_{i=1}^n$. Similar notations are used for sequences of matrices $\{\mathbf{X}_i\}_{i=1}^l$ and vectors $\{\mathbf{x}_i\}_{i=1}^l$, whose entries are indexed with convention $\mathbf{X}_i = [\mathbf{X}_i]_{m,n}$ and $\mathbf{x}_i = (x_{i,1}, x_{i,2}, \ldots, x_{i,n})$, respectively. The set of integers $\{m, \ldots, n\}$ is denoted by $[\![m, n]\!]$. We denote the $\ell_p$ (semi-)norm of a vector $\mathbf{x} \in \mathbb{R}^d$ as $\|\mathbf{x}\|_p = (\sum_{i=1}^d |x_i|^p)^{1/p}$ for all $p \in (0, \infty)$ and $\|\mathbf{x}\|$ implies $\|\mathbf{x}\|_2$. For a matrix $\mathbf{A} \in \mathbb{R}^{n \times m}$, $\|\mathbf{A}\|$ denotes its Frobenius norm.

**Fully connected neural networks.** In the entirety of the paper, we consider a multi-class classification setting. We denote the space of data points by $\mathcal{Z} = \mathcal{X} \times \mathcal{Y}$, where $\mathcal{X} \subset \mathbb{R}^{d_X}$ is the space of features and $\mathcal{Y} = \{1, 2, \ldots, d_Y\}$ is the space of the labels. Similar to prior art (e.g., [NBS18]), we focus our attention on the bounded feature domain $\mathcal{X} = \mathcal{X}_B := \{\mathbf{x} \in \mathbb{R}^{d_X}; \|\mathbf{x}\| \le B\}$.

We denote a *fully connected neural network* with $L$ layers by a collection of *weight matrices* $\{\mathbf{W}_l\}_{l=1}^L$, such that $\mathbf{W}_l \in \mathbb{R}^{h_l \times h_{l-1}}$, where $h_l$ denotes the number of hidden units at layer $l$ with $h_0 = d_X$ and $h_L = d_Y$. Accordingly, the *prediction function* $f_{\mathbf{w}}(\mathbf{x}) : \mathcal{X}_B \mapsto \mathbb{R}^{d_Y}$, with elements $f_{\mathbf{w}}(\mathbf{x}) := (f_{\mathbf{w}}(\mathbf{x})[1], \ldots, f_{\mathbf{w}}(\mathbf{x})[d_Y])$, corresponding to the neural network is defined as follows:

$$f_{\mathbf{w}}(\mathbf{x}) = \mathbf{W}_L \phi(\mathbf{W}_{L-1} \phi(\cdots \phi(\mathbf{W}_1 \mathbf{x}))), \tag{1}$$

where $\phi : \mathbb{R} \to \mathbb{R}$ is the rectified linear unit (ReLU) activation function, *i.e.*, $\phi(x) = \max(0, x)$, and it is applied element-wise when its input is a vector. For notational convenience, let us define $d_l := h_l \times h_{l-1}$ and $d := \sum_{l=1}^L d_l$. Furthermore, let $\mathbf{w}_l$ denote the *vectorized* weight matrix of layer $l$, *i.e.*, $\mathbf{w}_l := \mathbf{vec}(\mathbf{W}_l) \in \mathbb{R}^{d_l}$, where $\mathbf{vec}$ denotes vectorization. Finally, let $\mathbf{w}$ be the concatenation of all the vectorized weight matrices, i.e., $\mathbf{w} := [\mathbf{w}_1, \ldots, \mathbf{w}_L] \in \mathbb{R}^d$. We assume that $L \geq 2$.

**Risk minimization and SGD.** In order to assess the quality of a neural network represented by its weights $\mathbf{w}$, we consider a loss function $\ell : \mathcal{Y} \times \mathcal{Y} \mapsto \mathbb{R}_+$, such that $\ell(y, f_{\mathbf{w}}(\mathbf{x}))$ measures the loss incurred by predicting the label of $\mathbf{x}$ as $\arg\max_j f_{\mathbf{w}}(\mathbf{x})[j]$, when the true label is $y$. By following a standard statistical learning theoretical setup, we consider an unknown *data distribution* $\mu_Z$ over $\mathcal{Z}$, and a *training dataset* with $n$ elements, *i.e.*, $S = \{\mathbf{z}_1, \ldots, \mathbf{z}_n\}$, where each $\mathbf{z}_i =: (\mathbf{x}_i, y_i) \overset{\text{i.i.d.}}{\sim} \mu_Z$. We then denote the *population* and *empirical risks* as $\mathcal{R}(\mathbf{w}) := \mathbb{E}_{(\mathbf{x},y) \sim \mu_Z}[\ell(y, f_{\mathbf{w}}(\mathbf{x}))]$ and $\widehat{\mathcal{R}}(\mathbf{w}) := \frac{1}{n} \sum_{i=1}^n \ell(y_i, f_{\mathbf{w}}(\mathbf{x}_i))$.

Since $\mu_Z$ is unknown, we cannot directly attempt to minimize $\mathcal{R}$ in practice. One popular approach to address this problem is the *empirical risk minimization* strategy, where the goal is to solve the following optimization problem: $\min_{\mathbf{w} \in \mathbb{R}^d} \widehat{\mathcal{R}}(\mathbf{w})$. To tackle this problem, SGD has been one of the most popular optimization algorithms, which is based on the following simple recursion:

$$\mathbf{w}^{\{k+1\}} = \mathbf{w}^{\{k\}} - \eta \nabla \widetilde{\mathcal{R}}_{k+1}(\mathbf{w}^{\{k\}}), \text{ where } \nabla \widetilde{\mathcal{R}}_k(\mathbf{w}) := (1/b) \sum_{i \in \Omega_k} \nabla \ell(y_i, f_{\mathbf{w}}(\mathbf{x}_i)). \tag{2}$$

Here, $\mathbf{w}^{\{k\}}$ denotes the weight vector at iteration $k \in \mathbb{N}^+$, $\eta > 0$ is the step-size (or learning rate), $\nabla \widetilde{\mathcal{R}}_k(\cdot)$ is the stochastic gradient, and $\Omega_k$ is the mini-batch with size $b$, drawn with or without replacement from $[\![1, n]\!]$.

**Heavy-tailed distributions and the $\alpha$-stable family.** In this study, we will mainly deal with heavy-tailed random variables. While there exist different definitions of heavy tails in the literature, here, we call a random variable heavy-tailed if its distribution function has a *power-law* decay, *i.e.*, $\mathbb{P}(X \geq x) \sim cx^{-\alpha}$ as $x \to \infty$, for some $c > 0$ and $\alpha \in (0, 2)$. Here, the *tail index* $\alpha$ determines the tail thickness of the distribution, *i.e.*, as $\alpha$ get smaller, the distribution becomes heavier-tailed.

An important subclass of heavy-tailed distributions is the family of stable distributions. A random variable $X$ has symmetric $\alpha$-stable distribution, denoted by $\mathcal{S}\alpha\mathcal{S}(\sigma)$, if its characteristic function is equal to $\mathbb{E}[\exp(iwX)] = \exp(-|\sigma w|^\alpha)$, where $\alpha \in (0, 2]$ is again the tail index and $\sigma \in (0, \infty)$ is the scale parameter. An important property of $\mathcal{S}\alpha\mathcal{S}$ is that whenever $\alpha < 2$, $\mathbb{E}|X|^p$ is finite if and only if $p < \alpha$. This implies that the distribution has infinite variance as soon as $\alpha < 2$. In addition to their wide use in applied fields [Nol20], recently, $\mathcal{S}\alpha\mathcal{S}$ distributions have also been considered in deep learning theory [SSG19, PFF20, ZFM+20, ŞSDE20] and optimization [WGZ+21].

## 3 Compressibility and the Heavy-Tailed Mean-Field Regime

In this section, we will present our first set of theoretical results. We first identify a sufficient condition, then we prove that, under this condition, the compression errors of different pruning techniques become arbitrarily small as the network size increases.

### 3.1 The heavy-tailed mean-field regime

Due to the peculiar generalization behavior of neural networks trained with SGD, recent years have witnessed an extensive investigation of the theoretical properties of SGD in deep learning [Lon17, DR17, KL18, ZZL19, AZLL19, ŞSDE20, Neu21, CDE+21, BLGŞ21]. We will now mention two recently established theoretical properties of SGD, and our main contribution is studying the compressibility properties of the network under the assumption that these two seemingly unrelated 'phenomena' occur simultaneously.

***The heavy tail phenomenon.*** Several recent studies [MM19, SSG19, ŞGN+19, ŞZTG20, ZFM+20, ZKV+20, CWZ+21] have empirically illustrated that neural networks can exhibit heavy-tailed

behavior when optimized with SGD. Theoretically investigating the origins of this heavy-tailed behavior, [HM20] and [GŞZ21] later proved several theoretical results on online SGD, with rather surprising implications: due to the 'multiplicative' structure of the gradient noise (*i.e.*, $\nabla \widetilde{\mathcal{R}}_k(\mathbf{w}) - \nabla \widehat{\mathcal{R}}(\mathbf{w})$), as $k \to \infty$, the distribution of the SGD iterates $\mathbf{w}^{\{k\}}$ can converge to a heavy-tailed distribution with *infinite second-order moment*, and perhaps more surprisingly, this behavior can even emerge in simple linear regression with Gaussian data. The authors of [GŞZ21] further showed that, in the linear regression setting, the tail index $\alpha$ is *monotonic* with respect to the step-size $\eta$ and batch-size $b$: larger $\eta$ and/or smaller $b$ result in smaller $\alpha$ (*i.e.*, heavier tails).

***Propagation of chaos and mean-field limits.*** Another interesting property of SGD appears when the size of the network goes to infinity. Recently, it has been shown that, under an appropriate scaling of step-sizes, the empirical distribution of the network weights converges to a fixed *mean-field* distribution, and the SGD dynamics can be represented as a *gradient flow* in the space of probability measures [MMN18, CB18, SS20, DBDFŞ20]. Moreover, [DBDFŞ20] showed that a *propagation of chaos* phenomenon occurs in this setting, which indicates that when the network weights are initialized independently (*i.e.,* a-priori chaotic behavior), *the weights stay independent* as the algorithm evolves over time (*i.e.*, the chaos propagates) [Szn91].

Our focus in this paper is the setting where both of these phenomena occur *simultaneously*: we assume that as the size of the network goes to infinity, the SGD iterates become *independent* and their distribution converges to a *heavy-tailed* distribution. To formalize this setting, let us introduce the required notations. For an integer $m_l < d_l$, let $\mathbf{w}_{l,(1:m_l)}$ denote the first $m_l$ coordinates of $\mathbf{w}_l$, and for $m_l \geq d_l$ set $\mathbf{w}_{l,(1:m_l)} = \mathbf{w}_l$. Furthermore, parametrize the dimension of each layer $d_l$ with a parameter $\rho \in \mathbb{R}$, i.e., $d_l = d_l(\rho)$, such that for all $l = 1, \ldots, L$, we have:

$$d_l(\rho) \in \mathbb{N}_+, \quad \text{and} \quad \lim_{\rho \to \infty} d_l(\rho) = \infty. \tag{3}$$

This construction enables us to take the dimensions of each layer to infinity simultaneously [NP21].

**Condition 1** (Heavy-tailed mean-field limit condition)**.** *The SGD recursion* (2) *satisfies the heavy-tailed mean-field limit (HML) condition, if the dimensions obey* (3) *and if there exist heavy-tailed probability measures* $\{\mu_l^\star\}_{l=1}^L$ *on* $\mathbb{R}$ *with tail indices* $\{\alpha_l\}_{l=1}^L$, *such that* $\mu_l^\star(\{0\}) = 0$ *for all $l$ and for any* $m_1, \ldots, m_L \in \mathbb{N}_+$, *the joint distribution of*

$$\left( \mathbf{w}_{1,(1:m_1)}^{\{k\}}, \ldots, \mathbf{w}_{L,(1:m_L)}^{\{k\}} \right) \quad \text{weakly converges to} \quad (\mu_1^\star)^{\otimes m_1} \otimes \cdots \otimes (\mu_L^\star)^{\otimes m_L}, \tag{4}$$

*as $\rho, k \to \infty$, where $\mu \otimes \nu$ denotes the product measure and $\mu^{\otimes n}$ denotes the $n$-fold product measure.*

Informally, the HML condition states that in the infinite size and infinite iteration limit, the entries of the weight matrices will become independent, and the distribution of the elements within the same layer will be identical and heavy-tailed. We acknowledge that the above condition may not be always satisfied, and theoretical evidence for this condition is provided in rather seemingly simpler settings such as linear regression [GŞZ21] and/or two-layer neural networks [DBDFŞ20]. However, we will empirically illustrate the behavior described by the HML condition in Section 5 in certain scenarios. Note that, in (4), the particular form of independence in the limit is not crucial and we will discuss weaker alternatives in Section 3.3.

We further note that [DBDFŞ20] proved that a similar form of (4) indeed holds for SGD applied on single hidden-layered neural networks, where the limiting distributions possess second-order moments (*i.e.*, not heavy-tailed) and the independence is column-wise. Recently, [PFF20] investigated the infinite width limits of fully connected networks initialized from a $\mathcal{S}\alpha\mathcal{S}$ distribution and proved heavy-tailed limiting distributions. On the other hand, heavy-tailed propagation of chaos results have been proven in theoretical probability [JMW08, LMW20]; however, their connection to SGD has not been yet established. We believe that (4) can be shown to hold under appropriate conditions, which we leave as future work.

## 3.2 Analysis of compression algorithms

In this section, we will analyze the compression errors of three different compression schemes under the HML condition. All three methods are based on *pruning*, which we formally define as follows. Let $\mathbf{x}$ be a vector of length $d$, and consider its sorted version in descending order with respect to the magnitude of its entries, *i.e.*, $|x_{i_1}| \geq |x_{i_2}| \geq \cdots \geq |x_{i_d}|$, where $\{i_1, \ldots, i_d\} = \{1, \ldots, d\}$. For any $k \leq d$, the $k$-best term approximation of $\mathbf{x}$, denoted as $\mathbf{x}^{(k)} = (x_1^{(k)}, \ldots, x_d^{(k)})$, is defined as follows:

for $l \in [\![1, \lceil k \rceil ]\!]$, $x_{i_l}^{(k)} := x_{i_l}$ and for $l \notin [\![1, \lceil k \rceil ]\!]$, $x_{i_l}^{(k)} := 0$. Informally, we keep the $k$-largest entries of $\mathbf{x}$ with the largest magnitudes, and 'prune' the remaining ones. Current results pertain to one-shot pruning with no fine-tuning; other settings are left for future work (cf. [EKT20]).

In this section, we consider that we have access to a sample from the stationary distribution of the SGD, *i.e.* $\mathbf{w}^{\{\infty\}}$, and for conciseness we will simply denote it by $\mathbf{w}$. We then consider a *compressed network* $\widehat{\mathbf{w}}$ (that can be obtained by different compression schemes) and measure the performance of the compression scheme by its 'relative $\ell_p$-compression error' (cf. [AUM11, GCD12]), defined as: $\|\widehat{\mathbf{w}} - \mathbf{w}\|_p / \|\mathbf{w}\|_p$. Importantly for the following results, in the supplement, we further prove that a small compression error also implies a small perturbation on the network output.

***Magnitude pruning.*** Magnitude pruning has been one of the most common and efficient algorithms among all the network pruning strategies [HPTD15, BOFG20, KLG+21]. In this section, we consider the global and layer-wise magnitude pruning strategies under the HML condition.

More precisely, given a network weight vector $\mathbf{w} \in \mathbb{R}^d$ and a *remaining parameter ratio* $\kappa \in (0, 1)$, the global pruning strategy compresses $\mathbf{w}$ by using $\mathbf{w}^{(\kappa d)}$, *i.e.*, it prunes the smallest (in magnitude) $(1 - \kappa)d$ entries of $\mathbf{w}$. Also, note that $1/\kappa$ corresponds to the frequently used metric *compression rate* [BOFG20]. On the other hand, the layer-wise pruning strategy applies the same approach to each layer separately, *i.e.*, given layer-wise remaining parameter ratios $\kappa_l \in (0, 1)$, we compress each layer weight $\mathbf{w}_l \in \mathbb{R}^{d_l}$ by using $\mathbf{w}_l^{(\kappa_l d_l)}$. The following result shows that the compression error of magnitude pruning can be made arbitrarily small as the network size grows.

**Theorem 1.** *Assume that the recursion* (2) *satisfies the HML condition.*

*(i)* *Global magnitude pruning:* *if the weights of all layers have identical asymptotic distributions* $\mu_l^\star \equiv \mu^\star$ *with tail index* $\alpha_l^\star = \alpha$, *for all* $l \in [\![1, L]\!]$, *then for every* $\epsilon > 0$, $\varepsilon > 0$, $\kappa \in (0, 1)$, *and* $p \geq \alpha$, *there exists* $d_0 \in \mathbb{N}$, *such that* $\|\mathbf{w}^{(\kappa d)} - \mathbf{w}\|_p \leq \varepsilon \|\mathbf{w}\|_p$ *holds with probability at least* $1 - \epsilon$, *for* $d \geq d_0$.

*(ii)* *Layer-wise magnitude pruning:* *for every* $\epsilon > 0$, $\varepsilon_l > 0$ *and* $\kappa_l \in (0, 1)$, *where* $l \in [\![1, L]\!]$, *and* $p \geq \max_l \alpha_l$, *there exists* $d_{l,0} \in \mathbb{N}$, *such that* $\|\mathbf{w}_l^{(\kappa_l d_l)} - \mathbf{w}_l\|_p \leq \varepsilon_l \|\mathbf{w}_l\|_p$ *holds with probability at least* $1 - \epsilon$, *for* $d_l \geq d_{l,0}$.

This result shows that any pruning ratio of the weights is achievable as long as the network size is sufficiently large and the network weights are close enough to an i.i.d. heavy-tailed distribution. Empirical studies report that global magnitude pruning often works better than layer-wise magnitude pruning [BOFG20], except when it leads to over-aggressive pruning of particular layers [WZG20].

The success of this strategy under the HML condition is due to a result from compressed sensing theory, concurrently proven in [GCD12, AUM11], which informally states that for a large vector of i.i.d. heavy-tailed random variables, the norm of the vector is mainly determined by a small fraction of its entries. We also illustrate this visually in the supplementary document.

An important question here is that, to achieve a fixed relative error $\varepsilon_l$, how would the smallest $\kappa_l$ differ with varying tail-indices $\alpha_l$, *i.e.*, whether "heavier tails imply more compression". In our experiments, we illustrate this behavior positively: heavier-tailed weights are indeed more compressible. We partly justify this behavior for a certain range of $p$ in the supplement; however, a more comprehensive theory is needed, which we leave as future work.

***Singular value pruning.*** In recent studies, it has been illustrated that the magnitudes of the eigenvalues of the sample covariance matrices (for different layers) can decay quickly, hence pruning the singular values of the weight matrices, *i.e.*, only keeping the largest singular values and corresponding singular vectors, is a sensible pruning method. Exploiting the low-rank nature of fully connected and convolutional layer weights in network compression has been investigated theoretically and empirically [AGNZ18, YLWT17]. Here we will present a simple scheme to demonstrate our results.

More precisely, for the weight matrix at layer $l$, $\mathbf{W}_l$, consider its singular value decomposition, $\mathbf{W}_l = \mathbf{U}\mathbf{\Sigma}\mathbf{V}^\top$, and then, with a slight abuse of notation, define $\mathbf{W}_l^{[\kappa_l h_{l-1}]} := \mathbf{U}\mathbf{\Sigma}^{(\kappa_l h_{l-1})}\mathbf{V}^\top$, where $\mathbf{\Sigma}^{(\kappa_l h_{l-1})}$ is the diagonal matrix whose diagonal entries contain the $\lceil \kappa_l h_{l-1} \rceil$ largest singular values (*i.e.*, prune the diagonal of $\mathbf{\Sigma}$). Accordingly, denote $\mathbf{w}_l^{[\kappa_l h_{l-1}]} := \mathbf{vec}(\mathbf{W}_l^{[\kappa_l h_{l-1}]})$.

The next theorem shows that under the HML condition with an additional requirement that the limiting distributions are $\mathcal{S}\alpha\mathcal{S}$, the eigenvalues of the (properly normalized) sample covariance matrices will be indeed compressible and the pruning strategy achieves negligible errors as the network size grows.

**Theorem 2.** *Assume that the recursion* (2) *satisfies the HML condition, $L \geq 3$, and for all $l \in [\![2, L-1]\!]$, $\mu_l^\star \equiv \mathcal{S}\alpha_l\mathcal{S}(\sigma_l)$ with some $\sigma_l > 0$. Then, for every $\epsilon > 0$, $\varepsilon_l > 0$, and $\kappa_l \in (0, 1)$, there exists $\{h_{l,0}\}_{l=1}^{L}$, such that the following inequalities hold for every $h_l \geq h_{l,0}$ and $p \geq \max_l \alpha_l/2$:*

$$\|\boldsymbol{\lambda}_l^{(\kappa_l h_{l-1})} - \boldsymbol{\lambda}_l\|_p \leq \varepsilon_l^2 \|\boldsymbol{\lambda}_l\|_p, \quad and \quad \|\mathbf{w}_l^{[\kappa_l h_{l-1}]} - \mathbf{w}_l\| \leq \varepsilon_l \|\mathbf{w}_l\|, \tag{5}$$

*with probability at least $1 - \epsilon$, where $\boldsymbol{\lambda}_l \in \mathbb{R}^{h_{l-1}}$ denotes the vector of eigenvalues corresponding to the sample covariance matrix $h_l^{-2/\alpha_l} \mathbf{W}_l^\top \mathbf{W}_l$.*

We shall note that the proof of Theorem 2 in fact only requires the limiting distributions to be 'regularly varying' [TTR$^+$20] and symmetric around zero, which covers a broad range of heavy-tailed distributions beyond the $\alpha$-stable family [BDM$^+$16]. The sole reason why we require the $\mathcal{S}\alpha\mathcal{S}$ condition here is to avoid introducing further technical notation.

In Theorem 2, $p$ needs to be greater than or equal to $\alpha/2$, in contrast to the condition $p \geq \alpha$ in Theorem 1. The reason is that Theorem 2 is based on pruning the eigenvalues of the normalized covariance matrix and moreover in general if a matrix $\mathbf{A} \in \mathbb{R}^{n \times m}$ has elements $[\mathbf{A}]_{i,j}$ that are independent and identically distributed from the symmetric $\alpha$-stable distribution, then by [TTR$^+$20, Theorem 2.7], as $m \to \infty$, the eigenvalues of $m^{-2/\alpha}\mathbf{A}^T\mathbf{A}$ weakly converge to independent random variables, that are identically distributed from a positive stable distribution with tail index $\alpha/2$.

***Node pruning.*** The last pruning strategy that we consider is a structured pruning strategy, that corresponds to the removal of the whole columns of a fully connected layer weight matrix. Even though below we consider pruning based on the norms of the weight layer columns, the same arguments apply for pruning rows; see supplement for further discussion.

The idea in column pruning is that, for a given layer $l$, we first sort the columns of the weight matrix $\mathbf{W}_l \in \mathbb{R}^{h_l \times h_{l-1}}$ with respect to their $\ell_p$-norms for a given $p \geq \max_l \alpha_l$. Then, we remove the *entire columns* that have the smallest $\ell_p$-norms. More precisely, let $\mathbf{W}_l(i) \in \mathbb{R}^{h_l}$ be the $i$-th column of $\mathbf{W}_l$, for $i \in [\![1, h_{l-1}]\!]$, and suppose that $\|\mathbf{W}_l(i_1)\|_p \geq \|\mathbf{W}_l(i_2)\|_p \geq \cdots \geq \|\mathbf{W}_l(i_{h_{l-1}})\|_p$, where $\{i_1, \ldots, i_{h_{l-1}}\} = \{1, \ldots, h_{l-1}\}$. Then, we define the $k$-best column approximation of $\mathbf{W}_l$, denoted as $\mathbf{W}_l^{\{k\},p} \in \mathbb{R}^{h_l \times h_{l-1}}$, as follows: for $j \in [\![1, \lceil k \rceil]\!]$, $\mathbf{W}_l^{\{k\},p}(i_j) := \mathbf{W}_l(i_j)$ and for $j \notin [\![1, \lceil k \rceil]\!]$, $\mathbf{W}_l^{\{k\},p}(i_j) := 0$. Denote also $\mathbf{w}_l^{\{k\},p} := \mathbf{vec}\left(\mathbf{W}_l^{\{k\},p}\right)$.

**Theorem 3.** *Assume that the recursion* (2) *satisfies the HML condition. Then, for every $\epsilon > 0$, $\varepsilon_l > 0$, $\kappa_l \in (0, 1)$, where $l \in [\![2, L]\!]$, and $p \geq \max_l \alpha_l$, there exists $h_{l-1,0} \in \mathbb{N}$, such that $\|\mathbf{w}_l^{\{\kappa_l h_{l-1}\},p} - \mathbf{w}_l\|_p \leq \varepsilon_l \|\mathbf{w}_l\|_p$ holds with probability at least $1 - \epsilon$, for every $h_{l-1} \geq h_{l-1,0}$.*

This theorem indicates that we can remove entire columns in each layer, without considerably affecting the network weights, as long as the network is large enough. In other words, effectively the widths of the layers can be reduced. Structured pruning schemes are commonly used in CNNs where filters, channels, or kernels can be pruned by norm-based or other criteria [LKD$^+$17, HZS17, HPTD15].

### 3.3 A note on the limiting independence structure in the HML condition

We conclude this section by discussing the particular independence condition in the limit, which appears in Condition 1. We shall underline that the element-wise independence is not a necessity and under weaker conditions, we can still obtain Theorems 1-3 with identical or almost identical proofs. For instance, the proof of Theorem 3 remains the same when the columns of $\mathbf{W}_l$ are i.i.d. vectors with *dependent* components; hence, the element-wise independence is indeed not needed, but is used for the clarity of presentation. More generally, in all three theorems, the main requirement is to ensure a *weak dependence* between the components of the weight vector. More precisely, for Theorems 1, 2, 3, we respectively need (1) the entries of $\mathbf{W}_l$ or (2) its singular values, or (3) the $\ell_p$-norms of its columns to be *stationary* and *ergodic* with a heavy-tailed distribution. Under this condition, the same proof strategies will still work by invoking [SD15, Theorem 1], instead of [GCD12, Proposition 1].

# 4 Generalization Bounds

So far, we have shown that the heavy-tailed behavior in the weights of a network together with their independence result in compressibility. In this section, we further show that these phenomena bring forth a better generalization performance bound in the network. More precisely, we establish a generalization bound such that if a network is more compressible, then the corresponding compressed network has a smaller generalization error bound. Throughout the section, we will focus on layer-wise magnitude pruning; yet, our results can be easily extended to the other pruning strategies. Note that similar results have already been proven in [AGNZ18, SAM+20, SAN20, HJTW21]; yet, they cannot be directly used in our specific setting, hence, we prove results that are customized to our setup. More precisely, although these works, similar to our result, are based on the assumption of the "compressibility" of the network and moreover the generalization gap of the population risk with respect to the empirical margin risk is considered in [AGNZ18, HJTW21], they cannot be applied directly to the $\ell_p$-compressibility based strategies discussed in the previous section. We should emphasize that our results, as well as the previous mentioned works, are merely *upper bounds* on the generalization performance of the network.

Our generalization bounds are derived by applying the previously developed techniques as in [AGNZ18, NBS18]. In particular, we follow the approach of [AGNZ18] by further adapting the technique for the magnitude pruning strategy and allowing the compressed network weights to take unbounded continuous values. As in [AGNZ18], we consider the 0-1 loss function with *margin* $\gamma \geq 0$, $\ell_\gamma \colon \mathcal{Y} \times \mathcal{Y} \mapsto \{0, 1\}$, for the multiclass classifier $f_{\mathbf{w}}$, given as follows:

$$\ell_\gamma(y, f_{\mathbf{w}}(\mathbf{x})) = \begin{cases} 1, & \text{if } f_{\mathbf{w}}(\mathbf{x})[y] - \max_{j \neq y} f_{\mathbf{w}}(\mathbf{x})[j] \leq \gamma, \\ 0, & \text{otherwise.} \end{cases} \tag{6}$$

Still denoting $\mathbf{w} = \mathbf{w}^{\{\infty\}}$, the population and empirical risks associated with $\ell_\gamma$ are denoted as $\mathcal{R}_\gamma(\mathbf{w})$ and $\widehat{\mathcal{R}}_\gamma(\mathbf{w})$, respectively. By having the dataset $S \sim \mu_Z^{\otimes n}$, we assume the weights $\mathbf{w}$ are sampled according to the stationary distribution $P_{\mathbf{w}|S}$ of SGD. Denote the joint distribution of $(S, \mathbf{w})$ by $P_{S,\mathbf{w}} := \mu_Z^{\otimes n} P_{\mathbf{w}|S}$.

In the following theoretical results, we will assume that we have access to a random compressible neural network that is amenable to the layer-wise magnitude pruning strategy with high probability. This assumption is essentially the outcome of Theorem 1 under the HML condition together with an additional uniformity requirement on $d_{l,0}$ over $S$[1].

**H 1.** *For $\varepsilon \geq 0$, $\epsilon > 0$, and $\{\kappa_l\}_{l=1}^L \colon \kappa_l \in (0, 1)$, there exists $\{d_{l,0}\}_{l=1}^L \colon d_{l,0} \in \mathbb{N}$ independent of $S$, such that for $d_l \geq d_{l,0}$, $l \in [\![1, L]\!]$, the relation $\|\widehat{\mathbf{w}}_l - \mathbf{w}_l\| \leq \varepsilon \|\mathbf{w}_l\|$ holds for all $l \in [\![1, L]\!]$ simultaneously with probability at least $1 - \epsilon$, where $\widehat{\mathbf{w}}_l := \mathbf{w}_l^{(\kappa_l d_l)}$ and the probability is w.r.t. $P_{S,\mathbf{w}}$.*

In the following result, we relate the population risk of the compressed network to the empirical margin risk of the original (uncompressed) network. For notational convenience, for $\delta, \tau > 0$, let

$$R(\delta) := \inf\{R \colon \mathbb{P}(\|\mathbf{w}\| \geq R) \leq \delta\}, \text{ and } \mathcal{L}(\tau, \delta) := \sqrt{2} BL \Big(2R(\delta)/\sqrt{L}\Big)^{L-1}/\tau, \tag{7}$$

where the probability is with respect to the joint distribution $P_{S,\mathbf{w}}$.

**Theorem 4.** *Assume H1 holds. Then for $n \colon n/\log(n) \geq 10L$, $\{d_l\}_{l=1}^L \colon d_l \geq d_{l,0}$, and any $\delta, \tau > 0$, with probability at least $1 - 2e^{-\kappa d/2} - \delta - \epsilon$,*

$$\mathcal{R}_0(\widehat{\mathbf{w}}) \leq \widehat{\mathcal{R}}_{\gamma(\delta,\tau)}(\mathbf{w}) + \Big(12\mathcal{L}(\tau, \delta) R(\delta) + \sqrt{d}\Big)\sqrt{(\kappa + \epsilon_\kappa) \log(n)/n}, \tag{8}$$

*where $R(\delta)$ and $\mathcal{L}(\tau, \delta)$ are defined in (7), $\kappa := \frac{1}{d} \sum_{l=1}^L \lceil \kappa_l d_l \rceil$,[2]*

$$\epsilon_\kappa := (2h_b(\kappa) - \kappa \log(\kappa))/\log(n), \text{ and } \gamma(\delta, \tau) := \tau + \frac{\sqrt{2}B}{\tau}\Big(R(\delta)/\sqrt{L}\Big)^L\big((1 + \varepsilon)^L - 1\big).$$

---

[1]Note that the uniformity assumption is mild and can be avoided by combining Theorem 1 with Egoroff's theorem with additional effort, as in [ŞSDE20].

[2]The binary entropy $h_b(\kappa)$ (in nats) is defined as $-\kappa \log(\kappa) - (1 - \kappa)\log(1 - \kappa)$ for $\kappa \in [0, 1]$, with the convention $h_b(0) = h_b(1) = 0$. Note that $0 \leq h_b(\kappa) \leq \log(2)$.

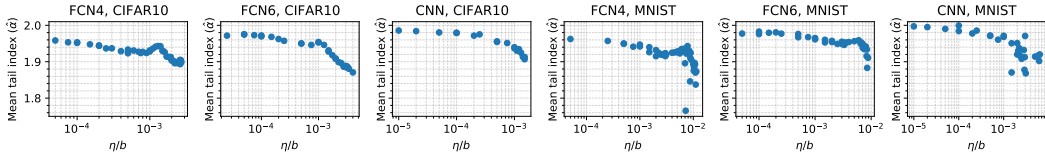

Figure 1: Mean estimated tail index ($\widehat{\alpha}$) vs. $\eta/b$ for each trained model. $x$-axes are log-scaled.

This result establishes a generalization bound such that, for a fixed relative compression error $\varepsilon$, if a network is more compressible, *i.e.* if $\kappa$ is smaller, then its corresponding compressed network has a smaller generalization error bound, as the bound is scaled by the factor $\approx \sqrt{\kappa}$.

In our proof, we prove an intermediate result, inspired by [NTS15] and stated in the supplement, which bounds the perturbation of the network output after compression. This guarantees that the risks of the original and pruned networks can be made arbitrarily close, as long as the relative compression error is small enough.

To have a better understanding of the constants in Theorem 4, in the next result, we consider a special case where the weights follow a stable distribution, and we make the above bound more explicit.

**Corollary 1.** *Assume that for $l \in [\![1, L]\!]$ and $i \in [\![1, d_l]\!]$, the conditional distribution of $w_{l,i} \overset{i.i.d.}{\sim} \mathcal{S}\alpha_l\mathcal{S}(\sigma_l)$ with $\alpha_l \in (1, 2)$. Further, assume that the scale parameters satisfy the following property:*

$$\sigma^2 := \sum_{l=1}^{L} (d_l/d)(\sigma_l/\sigma_{\alpha_l})^2 = \left[ (4^{-1/\alpha}\sqrt{L}/3)\sigma_0 d^{-(1/2+1/\alpha)} \right]^2, \tag{9}$$

*where $\sigma_{\alpha_l} := (2\Gamma(-\alpha_l)\cos((2 - \alpha_l)\pi/2))^{1/\alpha_l}$, $\alpha := \min_l \alpha_l$, and $\sigma_0$ is a constant, and also $\{\alpha_l\}_{l=1}^{L}$ and $\sigma$ are independent from $S$. Then, for every $\varepsilon > 0$ and $\kappa_l \in (0, 1)$, $l \in [\![1, L]\!]$, there exists $d_{l,0} \in \mathbb{N}$, such that for $d_l \geq d_{l,0}$, $n: n/\log(n) \geq 10L$, and every $\tau > 0$, with probability at least $1 - 3d^{-\alpha/(2L)}$,*

$$\mathcal{R}_0(\widehat{\mathbf{w}}) \leq \hat{\mathcal{R}}_\gamma(\mathbf{w}) + \left( a\sigma_0^L/\tau + 1 \right)\sqrt{(\kappa + \epsilon_\kappa)d\log(n)/n}, \tag{10}$$

*where $\{\widehat{\mathbf{w}}_l\}_l = \{\mathbf{w}_l^{(\kappa_l d_l)}\}_l$, $\gamma := \tau + b_\varepsilon\sigma_0^L\sqrt{d}/\tau$, $a := 6\sqrt{2}B2^L L^{3/2}$, and $b_\varepsilon := \sqrt{2}B((1 + \varepsilon)^L - 1)$.*

To simplify our presentation, we have set the 'scale' of the distribution as a decreasing function of dimension, which intuitively states that the typical magnitude of each entry of the network will get smaller as the network grows. We observe that for a fixed $\varepsilon$ and $d$, the bound improves as pruning ratio, $1 - \kappa$, increases. This result is of interest in particular since it is observed experimentally (and in part, theoretically) that heavier-tailed weights are more compressible, and hence due to this result have better generalization bounds. This provides an alternative perspective to the recent bounds that aim at linking heavy tails to generalization through a geometric approach [ŞSDE20].

Finally, in the supplement, we further show that the uncompressed network also inherits this good generalization performance bound, which is consistent with the results of [HJTW21, KLG$^+$21]: if a network is more "compressible", not only the generalization performance for the compressed network but also for the original network improves. The generalization bound adapted to $\ell_p$- compressibility based strategies, discussed in the previous section, is highlighted in Section S2 of the supplement.

## 5 Experiments

In this section, we present experiments conducted with neural networks to investigate our theory. We use three different model architectures: a fully connected network with 4 hidden layers (FCN4), a fully connected network with 6 hidden layers (FCN6), and a convolutional neural network with 8 convolutional layers (CNN). Hidden layer widths were 2048 for both FCN models. All networks include ReLU activation functions and none include batch normalization, dropout, residual blocks, or any explicit regularization term in the loss function. Each model is trained on MNIST [LCB10] and CIFAR10 [Kri09] datasets under various hyperparameter settings, using the default splits for training and evaluation. The total number of parameters were approximately $14M$ for FCN4-MNIST, $19M$ for FCN4-CIFAR10, $23M$ for FCN6-MNIST, $27M$ for FCN6-CIFAR10, and $9M$ for both CNN models. All models were trained with SGD until convergence with constant learning rates and no momentum. The convergence criteria comprised $100\%$ training accuracy and a training negative log-likelihood

less than $5 \times 10^{-5}$. The training hyperparameter settings include two batch-sizes ($b = 50, 100$) and various learning rates ($\eta$) to generate a large range of $\eta/b$ values. See the supplement for more details.

By invoking [GŞZ21, Corollary 11] which shows the ergodic average of heavy-tailed SGD iterates converges to a multivariate stable distribution, after convergence, we kept running SGD for 1000 additional iterations to obtain the average of the parameters to be used in this estimation. The tail index estimations were made by the estimator proposed in [MMO15, Corollary 2.4], which has been used by other recent studies that estimate tail index in neural network parameters [ŞSDE20, GŞZ21, ZFM⁺20]. The ergodic averaging does not change the tail index of the parameters, and was employed to facilitate tail index estimation by enabling the utilization of the aforementioned estimator. The rest of the experiments have been conducted without ergodic averaging. We also observe that the results with/without ergodic averaging are virtually identical in both tail index estimation tasks and pruning experiments. In all pruning methods, the parameters were centered before pruning is conducted with the median value to conform with the tail index estimation.

**Training hyperparameters and layer statistics.** We first verify that models trained with higher learning rate to batch-size ratio ($\eta/b$) lead to heavier-tailed parameters, replicating the results presented in [GŞZ21]. For each trained model, we compute the mean of separately estimated tail indices for all layers, so that each model is represented by a single value. This averaging is a heuristic without a clear theoretical meaning, and has been also used by recent works in the literature [GŞZ21, MM19]. Results presented in Figure 1 demonstrates that higher $\eta/b$ leads to heavier tails (lower $\hat{\alpha}$).

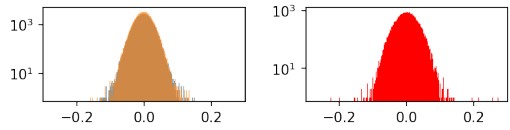

Figure 2: Empirical distribution of a CNN layer trained on MNIST. (Left) Overlaid histograms of a random partition of the weights, showing an identical distribution. (Right) Comparing the network weights to samples simulated i.i.d. from a symmetric $\alpha$-stable distribution with the same tail index $\alpha \approx 1.95$. $y$-axes are log-scaled.

Also of interest are the distribution of the resulting parameters from training. Figure 2 (left) demonstrates a representative example of the parameters from an CNN layer trained on MNIST, where two overlaid histograms representing empirical distributions of a random partition of the parameters are almost identical. On the right in the same figure, this distribution is compared to a simulated symmetric $\alpha$-stable distribution that has the same tail index ($\approx 1.95$). The figure demonstrates that the two distributions have similar qualitative properties.

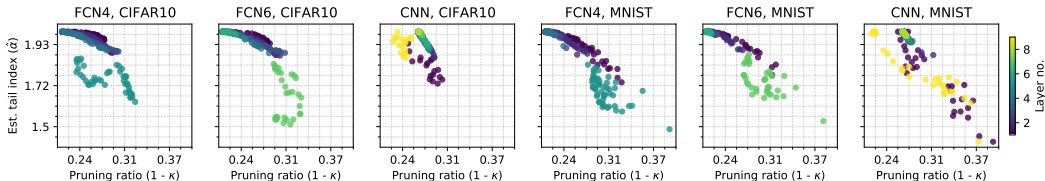

Figure 3: Estimated tail index ($\hat{\alpha}$) vs. pruning ratio ($1 - \kappa$), relative compression error $= 0.1$, $p = 2$

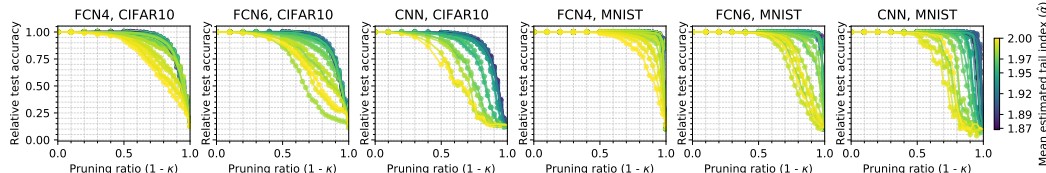

Figure 4: Relative test accuracy vs. pruning ratio for layer-wise magnitude pruning. Colors denote mean estimated tail index ($\hat{\alpha}$).

**Tail index and prunability.** In this section we examine whether networks with heavier-tailed layers, trained with higher $\eta/b$ ratios, are more prunable and whether they generalize better. As a baseline test, we first examine whether neural network layers which are heavier-tailed can be pruned more given a fixed maximum relative compression error. Figure 3 demonstrates for an error of $0.1$ that this is indeed the case. We next test our hypothesis that posit models with heavy-tailed parameters to be more prunable. Both the results pertaining to layer-wise magnitude pruning and singular value pruning, demonstrated in Figures 4 and 5 show that this is indeed the case. Here, relative test accuracy

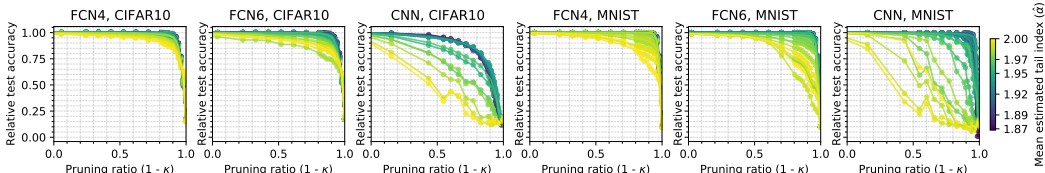

Figure 5: Relative test accuracy vs. pruning ratio for singular value pruning. Colors denote mean estimated tail index ($\hat{\alpha}$).

stands for test accuracy after pruning / unpruned test accuracy. The results show that models with heavier-tailed parameters (shown with darker colors) are starkly more robust against pruning. Similar results with global magnitude pruning can be seen in the supplement.

For structured pruning, we prune $3 \times 3$ kernel parameters in CNN models. The results (Figure 6) show a similar, hypothesis conforming pattern. Results for FCNs, presented in the supplement, were underwhelming; perhaps unsurprisingly as structured pruning is not as commonly used in FCNs [O'N20]. More successful attempts could be due to alternative scoring methods [SAM+20]; our approach might require wider layers to conform with theoretical conditions.

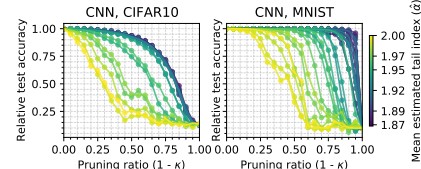

Figure 6: Relative test accuracy vs. pruning ratio for structured pruning. Colors denote mean estimated tail index ($\hat{\alpha}$).

**Tail index and generalization.** Following our theoretical results, we examine whether the heavier-tailed networks lead to better generalization performance. Consistent with our hypothesis, Figure 7 shows that models with the highest tail index have consistently the worst test performances. The same conclusion applies to generalization performance as training accuracy is $100\%$ for all models. See supplementary material for additional experiment results and discussion of the findings.

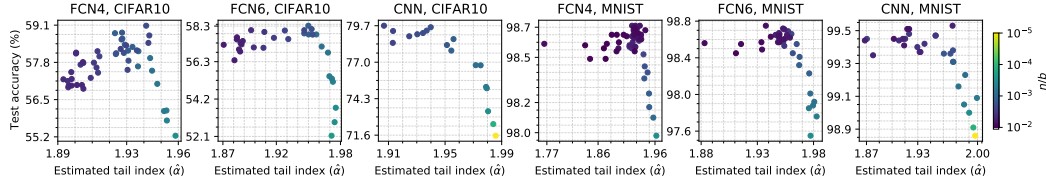

Figure 7: Test accuracy vs. mean estimated tail index ($\hat{\alpha}$) for each model. Color: training $\eta/b$ ratio.

## 6 Conclusion

We investigated the conditions under and the mechanism through which various pruning strategies can be expected to work, and confirmed the relationship between pruning and generalization from our theoretical approach. Future directions for research include formally identifying settings in which HML condition holds, eliciting the relationship between $\hat{\alpha}$ and $\kappa$ more clearly, and further examination of structured pruning in FCNs. Extending our work to other pruning strategies and network structures is also another important future direction. For example, it has recently been observed that gradients also exhibit heavy-tailed behavior [ŞGN+19, ZKV+20, ZFM+20]; we suspect that our theory might be applicable in the case of gradient-based pruning as well. The extension of our analyses to other network structures such as recurrent layers or ResNet blocks [HZRS15] would also be valuable.

## Acknowledgements

The authors are grateful to A. Taylan Cemgil, Valentin De Bortoli, and Mohsen Rezapour for fruitful discussions on various aspects of the paper. MAE is partially funded by CIFAR AI Chairs program, and CIFAR AI Catalyst grant, NSERC Grant [2019-06167]. UŞ's research is supported by the French government under management of Agence Nationale de la Recherche as part of the "Investissements d'avenir" program, reference ANR-19-P3IA-0001 (PRAIRIE 3IA Institute).

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
