# Heavy Tails in SGD and Compressibility of Overparametrized Neural Networks

## SUPPLEMENTARY DOCUMENT

This document provides the supplementary material associated with the NeurIPS 2021 paper entitled *"Heavy Tails in SGD and Compressibility of Overparametrized Neural Networks"*. We organize the document as follows:

- Section S1 describes the experimental setup used in our simulations, together with some additional experiment results and discussion.

- In Section S2, a generalization bound for an uncompressed network, given that this network is compressible, is presented.

- Section S3 provides an upper-bound on the change in the network output when there is a small change in the network weights.

- In Section S4, the relation between compressibility and the tail index is discussed.

- Proofs of the main results of the paper are presented in Section S5.

- Finally, the technical lemmas are proved in Section S6.

## S1 Details of the Experiments and Additional Results

Here we provide a more detailed explanation for our experimental setting, as well as the results and discussion we omitted from the main paper due to space restrictions.

### S1.1 Datasets

The experiments were conducted in a supervised learning setting where the task is classification of images. Each model is trained on CIFAR10 [Kri09] and MNIST [LCB10] datasets. The MNIST is an image classification dataset where the data is comprised of $28 \times 28$ black and white handwritten digits, belonging to one class from 0 to 9. We use the traditional split defined in the dataset where there are 60000 training and 10000 test samples. CIFAR10 is also image classification dataset comprising $32 \times 32$ color images of objects or animals, making up 10 classes. There are 50000 training and 10000 test images, this is the split that we use in the experiments.

### S1.2 Models

As described in the main text, in our experiments we use three models: a fully connected network with 4 hidden layers (FCN4), a fully connected network with 6 hidden layers (FCN6), and a convolutional neural network (CNN). Hidden layer widths are 2048 for the two FCN models. All networks include ReLU activation functions and none of them include batch normalization, dropout, residual layers, or any explicit regularization term in the loss function. The convolutional architecture for the CNN model for the CIFAR10 dataset progresses as below:

$$64, M, 128, M, 256, 256, M, 512, 512, M, 512, 512, M,$$

where integers stand for 2-dimensional convolutional layers (and the corresponding number of filters) with a kernel size of $3 \times 3$, and $M$ stands for $2 \times 2$ max-pooling with a stride of 2. Our CNN architecture follows that of VGG11 model [SZ15] except after the layers presented above we have only a single linear layer with a softmax output. For the MNIST experiment the first max-pooling layer was omitted as the dimensions of the MNIST images disallow the previous structure to be used. Table 1 includes the number of parameters for each model-dataset combination.

### S1.3 Training and hyperparameters

All models were trained with SGD until $100\%$ training accuracy and a training negative log-likelihood less than $5 \times 10^{-5}$ is acquired, with constant learning rates and no momentum. The training

|         | FCN4       | FCN6       | CNN       |
|---------|------------|------------|-----------|
| CIFAR10 | 18,894,848 | 27,283,456 | 9,222,848 |
| MNIST   | 14,209,024 | 22,597,632 | 9,221,696 |

Table 1: Parameter counts for all model-dataset combinations.

hyperparameters include two batch-sizes ($b = 50, 100$) and a variety of learning rates ($\eta$) to generate a large range of $\eta/b$ values. See the table below for the range of $\eta/b$ values created for each experiment setting. The ranges vary somewhat since different $\eta/b$ values might lead to heavy-tailed behavior (or divergence) under different settings. Table 2 presents these ranges for all experiments. See the source code for all experiment settings that were presented in the main results.

|         | FCN4 | FCN6 | CNN |
|---------|------|------|-----|
| CIFAR10 | $5 \times 10^{-5}$ to $2.7 \times 10^{-3}$ | $2.5 \times 10^{-5}$ to $4 \times 10^{-3}$ | $1 \times 10^{-5}$ to $1.5 \times 10^{-3}$ |
| MNIST   | $5 \times 10^{-5}$ to $1.14 \times 10^{-2}$ | $5 \times 10^{-5}$ to $8.8 \times 10^{-3}$ | $1 \times 10^{-5}$ to $6.35 \times 10^{-3}$ |

Table 2: $\eta/b$ ranges for all experiments.

## S1.4 Tail index estimation

We use the following multivariate tail index estimator proposed by [MMO15].

**Theorem S5** ([MMO15, Corollary 2.4]). *Let $\{X_i\}_{i=1}^{K}$ be a collection of i.i.d. random vectors where each $X_i$ is multivariate strictly stable with tail index $\alpha$, and $K = K_1 \times K_2$. Define $Y_i \triangleq \sum_{j=1}^{K_1} X_{j+(i-1)K_1}$ for $i \in [\![1, K_2]\!]$. Then, the estimator*

$$\widehat{\frac{1}{\alpha}} \triangleq \frac{1}{\log K_1} \Big( \frac{1}{K_2} \sum_{i=1}^{K_2} \log \|Y_i\| - \frac{1}{K} \sum_{i=1}^{K} \log \|X_i\| \Big). \tag{S1}$$

*converges to $1/\alpha$ almost surely, as $K_2 \to \infty$.*

This estimator has been used in previous research such as [ŞGN$^+$19] and [TNT18]. We center the observations using the median values before the estimation. Using the alternative univariate tail index estimator [MMO15, Corollary 2.2] in the same paper has no qualitative effects on our results, an additional benefit of our choice is additional analyses it makes possible as presented in Section S1.6.2. Comparisons with alternative tail index estimators with symmetric $\alpha$-stable assumption revealed no dramatic differences between various estimators [SU20].

## S1.5 Pruning details

We first provide a review of the pruning methods we use. All three notions of pruning in our experiments correspond to the magnitude-wise ordering of certain parameters and the 'pruning' of a certain ratio of parameters that correspond to smallest magnitudes[3]. When the parameters that are pruned are the weight parameters themselves, this corresponds to magnitude-based pruning or *magnitude pruning* as known in the literature, which can be conducted layer-wise or globally [BOFG20]. *Singular value pruning*, as described here, corresponds to pruning of the smallest singular values (and, by implication, the related singular vectors) in the SVD of specific layers. To apply the SVD to CNN layers, we reshape the parameter tensors into matrices of shape (# channels) × (# filters ×3 × 3). Lastly *node pruning* corresponds to the pruning of the whole columns in the weight matrices. Again, CNN counterpart of node pruning is open to interpretation; we choose to prune specific kernels according to the their norms.

Before any pruning is done, the parameters to be pruned are centered with the estimated median of the observations, in order to conform with our tail index estimation methodology. We chose median due

---

[3]Note that a more relaxed definition of pruning would be 'systematic removal of model parameters' to allow for different scoring methods in pruning [BOFG20]. However, we proceed with our specific definition since this allows us to communicate our theoretical and experimental results more concisely.

to its robustness against extreme observations especially with a small sample - however our results were qualitatively unchanged when the mean was used in the centering. After the pruning (in all three methods), the median was added to the pruned parameter vectors before testing the performance of the resulting model. Note that the median (or mean) was usually very small in norm and omitting centering made no qualitative effect on the results.

Lastly, while 'remaining parameter ratio' ($\kappa$) or 'pruning ratio' ($1 - \kappa$) are easy to interpret in the case of magnitude pruning or node pruning, in SVD $\kappa$ would equal (number of singular value and vector parameters left) / (number of weight parameters in the layer), and pruning ratio would be determined accordingly.

### S1.6  Additional results and discussion

Here we present additional results and discussion that were referenced in the main text but were not presented due to space restrictions.

#### S1.6.1  Causal interpretation of the relationships in question

An appropriate question regarding our theoretical and experimental findings would be: Is a causal interpretation of the hypothesized relations warranted? Although the relationship among training hyperparameters, parameter tail index, compressibility, and generalization is inevitably multifaceted, we believe that there are grounds to interpret the relations causally in this context.

To be more specific, [GŞZ21, Theorem 4, Proposition 5] shows that the tail index is fully determined by $\eta$, $b$, and the Hessian of the problem in the context of quadratic optimization: the tail index changes monotonically with respect to both $\eta$ and $b$. In this paper we establish the relationship empirically in the context of neural networks, replicating the results presented in [GŞZ21]. We also show that the existence of heavy-tailed network parameters leads to compressibility (Theorems 1, 2, and 3), and thus to arbitrarily small perturbation in the network outputs when pruned (Lemma S1). We also demonstrate that the more compressible the network is, the smaller its generalization bound is (Theorem 4).

Using a different, geometric framework, [ŞSDE20, Appendix S1.2] (arxiv:2006.09313) experimentally demonstrated that a lower tail index leads to a better generalization, where they directly varied the tail index and monitored the generalization error, as the reviewer requested. Given these results and our experimental findings, we believe that a causal interpretation of the relationships in question is not without support.

**Investigation of tail index and prunability with synthetic data.**   Experimental manipulation of tail index directly in the case of neural networks trained with real data is hard to formulate and conduct. However, to examine this issue further, we conducted a number of experiments with synthetic data.

In this setting, we created neural networks that were structural analogues of FCN6 networks presented in the original experiments, that is, feedforward neural networks of 6 hidden layers and a width of 2048 units for each hidden layer, with ReLU activation functions after each hidden layer. For the experiments, for each $\alpha_i \in \{1.50, 1.55, 1.60, \ldots, 2.00\}$, we randomly sampled the parameters of these networks independently from a $\mathcal{S}\alpha\mathcal{S}$ distribution with an $\alpha_i$ tail index parameter. After the sampling of the layer parameters, each layer was converted to a unit length vector, in order to avoid the possibly confounding effects of scale between different tail indices. This procedure excluded the much smaller final layer, which was sampled as in the initialization of the original experiments.

We also created random data for these experiments, in the shape of the MNIST training data, sampled independently from a standard normal distribution. For each network, the labels for these data were created by passing the synthetic data through the synthetic network and choosing the label with the maximum final value. Then, for each network created, we conducted layer-wise magnitude pruning for different values of $\kappa$, and evaluated the performance of the pruned versions of the networks with their original performances. This was repeated 10 times, and Figure **??** presents the mean of these accuracy values for each tail index ($\alpha$) and pruning ratio ($1 - \kappa$) combination.

The results again support our hypothesis: networks with lower tail indices are more prunable, that is, more robust to pruning in terms of performance decay. This parallels the conclusion of the original experiments, where a similar conclusion was reached with networks trained on real data. We leave the

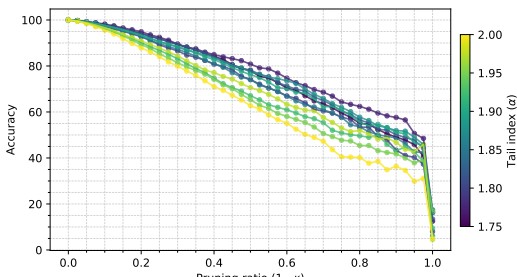

Figure S1: Accuracy ($\alpha$) vs. pruning ratio ($1 - \kappa$) for synthetically generated networks and data.

extension of these experiments for different data generation schemes, layer structures, and pruning types to future work.

### S1.6.2  Investigating the HML condition in synthetic experiments

Recall that in Figure 3 we examined, for a given relative compression error, whether lower tail index is associated with higher prunability. The results demonstrated that this was indeed the case. Here we compare our empirical results with some synthetic results to get additional insights regarding whether HML condition is actually observed in our networks.

For this experiment, we randomly sample tail indices $\alpha_i \sim \mathcal{U}(1.75, 2)$, where $i \in \{1, \ldots, 250\}$. Then for each $\alpha_i$, we sample three different 'weight matrices': $\mathbf{W}_{ind,i}, \mathbf{W}_{col,i}, \mathbf{W}_{lay,i} \in \mathbb{R}^{500 \times 500}$. The elements of $\mathbf{W}_{ind,i}$ are sampled independently from a $\mathcal{S}\alpha\mathcal{S}$ with tail index $\alpha_i$; this corresponds to the case where weight parameters are statistically independent as prescribed by the HML condition. On the other hand, columns of the $\mathbf{W}_{col,i}$ are independently sampled from a 500-dimensional multivariate *elliptically contoured* $\mathcal{S}\alpha\mathcal{S}$ with tail index $\alpha_i$. A $d$-dimensional elliptically contoured multivariate $\mathcal{S}\alpha\mathcal{S}$ has the characteristic function

$$\mathbb{E}[\exp(i\langle \omega, X \rangle)] = \exp(-\|\omega\|^{\alpha}),$$

where $X, \omega \in \mathbb{R}^d$ and $\langle \cdot, \cdot \rangle$ stands for inner product. This means that while the columns of the matrix are independent, column elements can be correlated. Lastly, all elements of $\mathbf{W}_{lay,i}$ are sampled from a $(500 \times 500)$-dimensional elliptically contoured multivariate $\mathcal{S}\alpha\mathcal{S}$, creating a case where all elements of the matrix can be correlated. We repeat the analysis presented in Figure 3 for all three sets of sampled synthetic layer weights, and present the results in Figure S2.

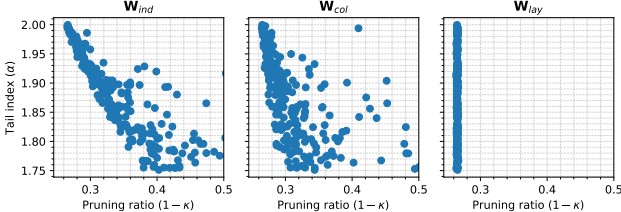

Figure S2: Tail index ($\alpha$) vs. pruning ratio ($1 - \kappa$), with relative compression error $= 0.1$, for synthetically generated weight matrices.

The results demonstrate two interesting phenomena. First, a comparison of these results with Figure 3 shows that our empirical results show the most similarity with results obtained with $\mathbf{W}_{ind,i}$, showing support for the existence of the HML condition. Another observation is that as the layer parameters become correlated, the prunability advantage conferred by heavier tails disappears. This observation both supports the existence of HML condition in overparametrized neural networks and invites further discussion on the importance of propagation of chaos phenomenon in such architectures for compressibility and generalization [DBDFŞ20].

### S1.6.3 Global magnitude pruning and node pruning results

In Figure S3 present the global magnitude pruning results for magnitude pruning. The results are qualitatively very similar to those of the layer-wise magnitude pruning. Figure S4 presents the results of node pruning on FCNs. As mentioned in the main text, the less impressive results might have to do with the layer widths not being sufficient for our theoretical conditions. A more favorable approach to structured pruning in FCNs would factor in the fact that removal of columns from a layer is also equivalent to the removal of corresponding rows from the previous layer. When computing the pruning ratio, factoring in these corresponding rows would produce more benevolent results. However, we have not done this in our experiments since this is not necessarily implied by our theory.

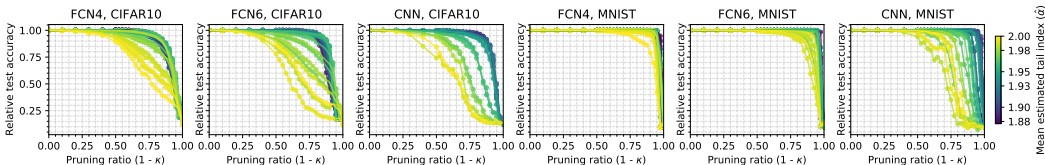

Figure S3: Relative test accuracy vs. pruning ratio for global magnitude pruning. Color: mean $\hat{\alpha}$.

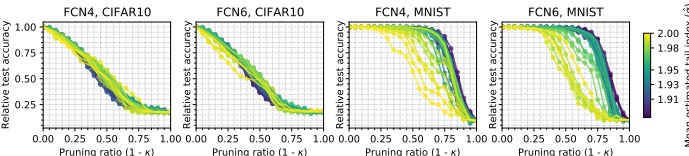

Figure S4: Relative test accuracy vs. pruning ratio for node pruning in FCNs. Color: mean $\hat{\alpha}$.

### S1.6.4 Experiments with ergodic averaged parameters

In the main article we note that whether the pruning experiments are conducted on the final parameters or the ergodic averaging thereof does not lead to any concrete differences in the results. The following Figures S5, S6, and S7, which include results obtained with parameters that were ergodically averaged, present results that were virtually identical with the originals prsesented in the paper.

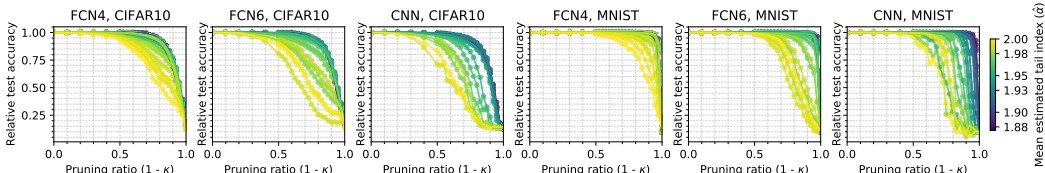

Figure S5: Relative test accuracy vs. pruning ratio for layer-wise magnitude pruning using ergodic averages. Color: mean $\hat{\alpha}$.

### S1.6.5 Experiments with CIFAR100

We also repeated our original experiments with the CIFAR100 dataset [Kri09]. Since the training error was much slower to decrease in this case, we changed our convergence criteria to a NLL of $< .01$ and an accuracy of $> 99\%$. Even though we were not able to allocate a comparable amount of resources to these set of experiments due to the total number of experiments that had to be conducted overall, the experiments we *were* able to conduct produced results firmly in the direction of our original experiments' results, further confirming our hypotheses. Figures S8, S9, S10, S11, and S12 presents these results in a form comparable to the originals.

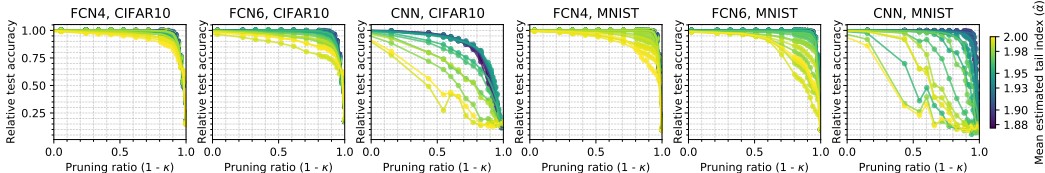

Figure S6: Relative test accuracy vs. pruning ratio for layer-wise singular value pruning using ergodic averages. Color: mean $\hat{\alpha}$.

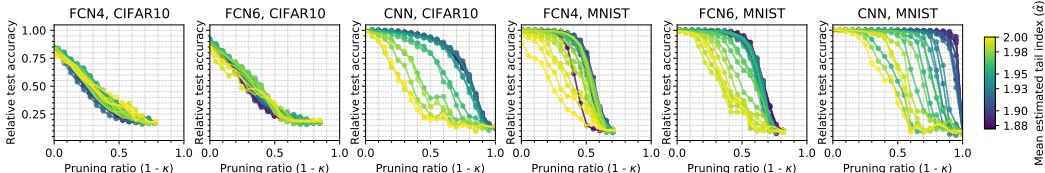

Figure S7: Relative test accuracy vs. pruning ratio for layer-wise node pruning using ergodic averages. Color: mean $\hat{\alpha}$.

## S1.7 Hardware and other resources

The experiments were conducted on an internal server of a research institute. Nvidia Titan X, 1080 Ti, and 1080 model GPU's were used roughly equally in running the experiments. Our published results involve 215 trained models, each of which included GPU-heavy workload, with an average completion time of approximately 2 hours. Around 30 models diverged (thus were not used in the results) and in most cases were trained for less than an hour. Total GPU-time could correspondingly be estimated to equal 460 hours. We also conducted tail index estimation and pruning experiments on these networks, however the computational load of these experiments are negligible compared to those of the training of the algorithms, with an estimated CPU time of 48 hours for all estimation and pruning tasks that were published.

As can be seen in the accompanying code, the experiments were conducted using Python programming language. The deep learning framework PyTorch [PGM$^+$19] as well as some of its tutorials[4] were used in implementing the experiments.

## S2 Generalization bound for the uncompressed network

In this section, in the continuation of Section 4, we establish a generalization bound for an uncompressed network, if this network is compressible using the layer-wise magnitude pruning strategy.

**Theorem S6.** *Assume* **H***1 holds. Then for* $n$: $n/\log(n) \geq \max(9L, 81\varepsilon^{-2\kappa})$, $\{d_l\}_{l=1}^L$: $d_l \geq d_{l,0}$ *and* $d \geq 10$*, and any* $\delta, \tau > 0$*, with probability at least* $1 - 2e^{-d/2} - \delta - \epsilon$,

$$\mathcal{R}_0(\mathbf{w}) \leq \widehat{\mathcal{R}}_\tau(\mathbf{w}) + \max\Big(2, 24\rho_\varepsilon(\kappa,d)\mathcal{L}(\tau,\delta)R(\delta)/\sqrt{d}\Big)\sqrt{d\log(n)/n}. \tag{S2}$$

*where* $R(\delta)$ *and* $\mathcal{L}(\tau,\delta)$ *are defined in* (7), $\kappa := \frac{1}{d}\sum_{l=1}^L \lceil \kappa_l d_l \rceil$,

$$\rho_\varepsilon(\kappa,d) := \min\Big(\varepsilon^{1-\kappa}\exp\Big(h_b(\kappa) + h_b^{(1)}(\kappa,d)\Big), 1\Big) \leq \min\big(3\varepsilon^{1-\kappa}, 1\big),$$

$$h_b^{(1)}(\kappa,d) := \frac{\lceil d/2\rceil}{d}\max(h_b(\lceil \kappa d/2\rceil/\lceil d/2\rceil), h_b(\lfloor \kappa d/2\rfloor/\lceil d/2\rceil)).$$

Note that the function $h_b^{(1)}(\kappa,d) \leq \log(2)(1/2 + 1/d)$. Hence $\varepsilon^{1-\kappa}e^{h_b(\kappa)+h_b^{(1)}(\kappa,d)} \leq \varepsilon^{1-\kappa}e^{log(2)(3/2+1/d)} \leq 3\varepsilon^{1-\kappa}$, for $d \geq 12$. Besides, $\rho_\varepsilon(1,d) = 1$ and $\rho_\varepsilon(0,d) = \varepsilon$. Moreover, when both $d$ and $\kappa d$ are even numbers, then $h_b^{(1)}(\kappa,d) = \frac{1}{2}h_b(\kappa)$ and $\rho_\varepsilon(\kappa) := \rho_\varepsilon(\kappa,d)$ is increasing with

---

[4]HTTPS://GITHUB.COM/PYTORCH/VISION/BLOB/MASTER/TORCHVISION/MODELS/VGG.PY

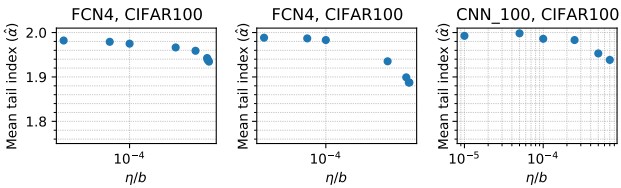

Figure S8: Mean estimated tail index ($\hat{\alpha}$) vs. $\eta/b$ for each trained model, using CIFAR100 dataset. $x$-axes are log-scaled.

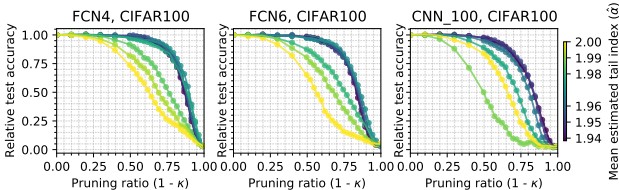

Figure S9: Relative test accuracy vs. pruning ratio for layer-wise magnitude pruning using ergodic averages, with CIFAR100 dataset. Color: mean $\hat{\alpha}$.

respect to $\kappa$. To show the latter claim, consider the derivative of $g(\kappa) := \varepsilon^{1-\kappa} \exp\left(\frac{3}{2}h_b(\kappa)\right)$ with respect to $\kappa$. This derivative is equal to zero at $\kappa^* = 1/\left(1 + \varepsilon^{2/3}\right)$ and is positive for $\kappa < \kappa^*$. In addition, $g(\kappa^*) = \left(1 + \varepsilon^{2/3}\right)^{3/2} > 1$. Hence, $\rho_\varepsilon(\kappa) = \min(g(\kappa), 1)$ is increasing with respect to $\kappa$. In Figure S13, $\rho_\varepsilon(\kappa) = \min\left(\varepsilon^{1-\kappa} \exp\left(\frac{3}{2}h_b(\kappa)\right), 1\right)$, together with its upper bound $\min\left(3\varepsilon^{1-\kappa}, 1\right)$ are plotted for different values of $\varepsilon$.

It can be observed that if a network is more compressible, then not only the compressed network, but also the original network has a better generalization bound. This result is consistent with the findings of [HJTW21, KLG⁺21]. In [HJTW21], it is shown that if two networks are "*close*" enough, a good generalization bound on one of them, would imply a good bound on the other one as well. In [KLG⁺21], it is shown that "*prunability*" of a network captures well the generalization property of the network.

Finally, it should be noted that when the weights of the network follow a stable distribution, similar results to Corollary 1 can be established for the original network.

## S3   Perturbation Bound

The goal of pruning is to find compressed weights $\widehat{\mathbf{w}}$ with low dimensionality that are close enough to the original weights $\mathbf{w}$, which is measured in this work by the relative error $\|\widehat{\mathbf{w}} - \mathbf{w}\|/\|\mathbf{w}\|$. The following perturbation result guarantees that such pruning strategies also result in small perturbations on the output of the network. The proof is based on the technique given in [NBS18].

**Lemma S1.** *Let* $\mathbf{w}, \widehat{\mathbf{w}} \in \mathbb{R}^d$ *be two fully connected neural networks. Assume that there exists* $\{\varepsilon_l\}_{l=1}^L : \varepsilon_l \geq 0$, *such that* $\|\widehat{\mathbf{w}}_l - \mathbf{w}_l\| \leq \varepsilon_l\|\mathbf{w}_l\|$, *for all* $l \in [\![1, L]\!]$. *Then, the following inequality holds:*

$$\|f_{\widehat{\mathbf{w}}}(\mathbf{x}) - f_{\mathbf{w}}(\mathbf{x})\| \leq B\left[\prod_{l=1}^L (1 + \varepsilon_l) - 1\right]\left[\prod_{l=1}^L \|\mathbf{w}_l\|\right], \tag{S3}$$

*for all* $\mathbf{x} \in \mathcal{X}_B$. *In particular, if* $\varepsilon_l = \varepsilon$ *for all layers and* $\|\mathbf{w}\| \leq R$, *then*

$$\|f_{\widehat{\mathbf{w}}}(\mathbf{x}) - f_{\mathbf{w}}(\mathbf{x})\| \leq B\left[(1 + \varepsilon)^L - 1\right]\left(R/\sqrt{L}\right)^L. \tag{S4}$$

For derivation of the above bound on the network outputs, the worst case in the propagation of the errors of each layer is assumed, which results into an exponential dependence on the depth of the network, similarly to [NBS18].

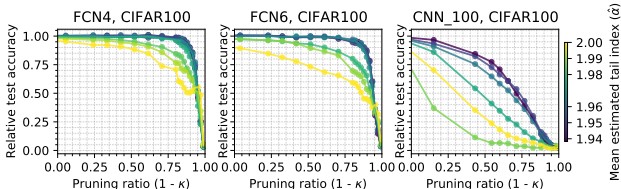

Figure S10: Relative test accuracy vs. pruning ratio for singular value pruning using ergodic averages, with CIFAR100 dataset. Color: mean $\hat{\alpha}$.

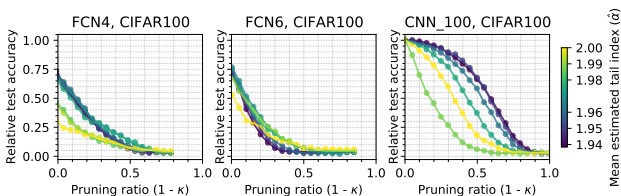

Figure S11: Relative test accuracy vs. pruning ratio for node pruning using ergodic averages, with CIFAR100 dataset. Color: mean $\hat{\alpha}$.

## S4  Compression rate and tail index

In this work, different pruning strategies have been investigated by exploiting the compressibility properties of heavy-tailed distributions. In this section, we show that moreover, in some certain sense, heavier-tailed distributions are more compressible. However, we must underline that this result is neither comprehensive, nor directly usable in our framework, as we will discuss after stating it.

Before stating the result, let us define the following quantity. For $\varepsilon > 0$ and $\mathbf{w} \in \mathbb{R}^d$, let

$$\kappa_p(\mathbf{w}, \varepsilon) \coloneqq \min\Big\{\kappa\colon \Big(\|\mathbf{w}^{(\kappa d)} - \mathbf{w}\|_p/\|\mathbf{w}\|_p\Big) \leq \varepsilon\Big\}. \tag{S5}$$

**Proposition S1.** *Suppose that $\mathbf{w}_1 \in \mathbb{R}^d$ and $\mathbf{w}_2 \in \mathbb{R}^d$ are independent vectors of i.i.d. heavy-tailed random variables with tail indices $\alpha_1$ and $\alpha_2$, respectively. If $\alpha_1 > \alpha_2$, then for any $\kappa, \varepsilon, \delta > 0$ and $p < \max(\alpha_1, \alpha_2)$, there exists $d_0(\delta)$, such that for $d \geq d_0(\delta)$,*

$$\mathbb{E}\Big[\|\mathbf{w}_1^{(\kappa d)} - \mathbf{w}_1\|_p/\|\mathbf{w}_1\|_p\Big] + \delta > \mathbb{E}\Big[\|\mathbf{w}_2^{(\kappa d)} - \mathbf{w}_2\|_p/\|\mathbf{w}_2\|_p\Big], \tag{S6}$$

*and*

$$\mathbb{E}[\kappa_p(\mathbf{w}_1, \varepsilon)] + \delta > \mathbb{E}[\kappa_p(\mathbf{w}_2, \varepsilon)]. \tag{S7}$$

The above proposition shows that for a fixed $p$-norm of the normalized compression error with $p < \alpha$, the heavier-tailed distributions are more compressible. The caveat here is that for $p \geq \max(\alpha_1, \alpha_2)$, all terms in (S6) and (S7) go to zero due to Lemma S2 and hence (S6) and (S7) trivially hold. Therefore, unfortunately we cannot use Proposition S1 in our framework since we are mainly interested in the case where $p \geq \max(\alpha_1, \alpha_2)$. Investigating the level of compressibility as a function of the tail index is a natural next step for our study.

## S5  Proofs of the Main Results

In this section we provide proofs of our main results. We shall begin with stating the following result from [GCD12], which will be repeatedly used in our proofs.

### S5.1  Existing Theoretical Results

Many of our results are based on the compressibility of i.i.d. instances of heavy-tailed random variables. Here, we state a known result regarding this fact.

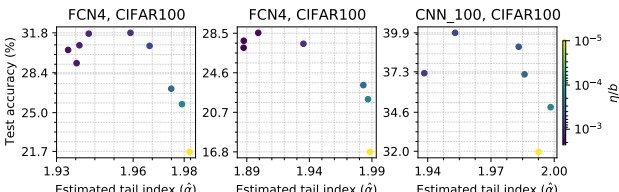

Figure S12: Test accuracy vs. mean estimated tail index ($\hat{\alpha}$) for each model, using CIFAR100 dataset. Color: training $\eta/b$ ratio.

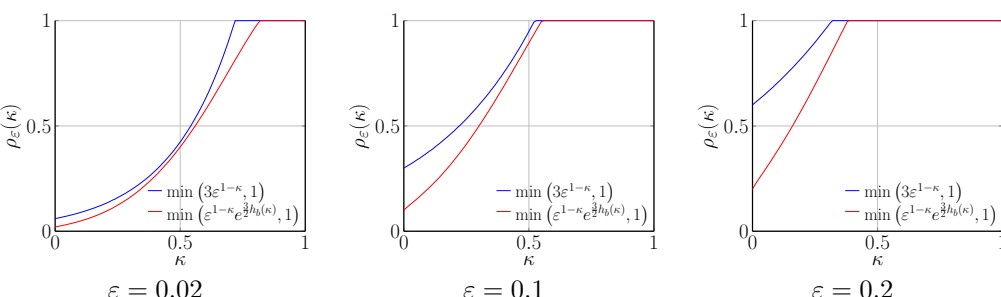

Figure S13: $\rho_\varepsilon(\kappa)$ for different values of $\varepsilon$.

**Lemma S2** ([GCD12, Proposition 1, Part 2] ). *Let $x \in \mathbb{R}$ be a random variable and assume that $\mathbb{E}|x|^\alpha = \infty$ for some $\alpha \in \mathbb{R}_+$. Then for all $p \geq \alpha$, $0 < \kappa \leq 1$ and any sequence $\kappa_d$ such that $\lim_{d \to \infty} \frac{\kappa_d}{d} = \kappa$, the following identity holds almost surely:*

$$\lim_{d \to \infty} \left( \|\mathbf{x}^{(\kappa_d)} - \mathbf{x}\|_p / \|\mathbf{x}\|_p \right) = 0, \tag{S8}$$

*where $\mathbf{x} = (x_1, \ldots, x_d)$ is a vector of i.i.d. random variables of length $d$.*

A similar result was concurrently proven in [AUM11]. As stated before, this result informally states that for a large vector of i.i.d. heavy-tailed random variables, the norm of the vector is mainly determined by a small fraction of its entries. To show this visually, we have generated $10^4$ i.i.d. random variables $\{x_i\}_i$ with $\mathcal{S}\alpha\mathcal{S}(1)$ distribution where $\alpha = 1.7$. Then, we have plotted the histogram of $|x|$ in Figure S14. As can be seen in the figure, the norm of the whole vector is mainly determined by few number of samples.

### S5.2 Proof of Theorem 1

*Proof.* (i)    As $\{d_l\}_{l=1}^L$ grow, due to HML condition and assumptions of this part of the theorem, $\mathbf{w}$ converges in distribution to a heavy-tailed random vector, denoted as $\mathbf{x} = (x_1, \ldots, x_d) \in \mathbb{R}^d$, with i.i.d. elements and tail index $\alpha \in (0, 2)$. Hence, for any $\epsilon > 0$, $\varepsilon > 0$, $\kappa \in (0, 1)$, and $p \geq \alpha$ there exists $\{d'_{l,0}\}_{l=1}^L : d'_{l,0} \in \mathbb{N}$ such that for $d_l \geq d'_{l,0}$, $l \in [\![1, L]\!]$,

$$\mathbb{P}\left[ \left( \|\mathbf{x}^{(\kappa d)} - \mathbf{x}\|_p / \|\mathbf{x}\|_p \right) \leq \varepsilon \right] - \mathbb{P}\left[ \left( \|\mathbf{w}^{(\kappa d)} - \mathbf{w}\|_p / \|\mathbf{w}\|_p \right) \leq \varepsilon \right] \leq \epsilon. \tag{S9}$$

Moreover due to Lemma S2, there exists $d''_0 \in \mathbb{N}$ such that for $d \geq d''_0$,

$$\mathbb{P}\left[ \left( \|\mathbf{x}^{(\kappa d)} - \mathbf{x}\|_p / \|\mathbf{x}\|_p \right) \leq \varepsilon \right] = 1 \tag{S10}$$

The results follows from (S9) and (S10) and by choosing $d_{l,0} \geq d'_{l,0}$, for $l \in [\![1, L]\!]$, such that $\sum_{l=1}^L d_{l,0} \geq d''_0$.

(ii)    The proof is similar to the previous part. As $\{d_l\}_{l=1}^L$ grow, due to HML condition, for $l \in [\![1, L]\!]$, $\mathbf{w}_l$ converges in distribution to a heavy-tailed random vector, denoted as $\mathbf{x}_l = (x_{l,1}, \ldots, x_{l,d_l}) \in \mathbb{R}^{d_l}$ , with i.i.d. elements and tail index $\alpha_l \in (0, 2)$. Hence, for any $\epsilon > 0$,

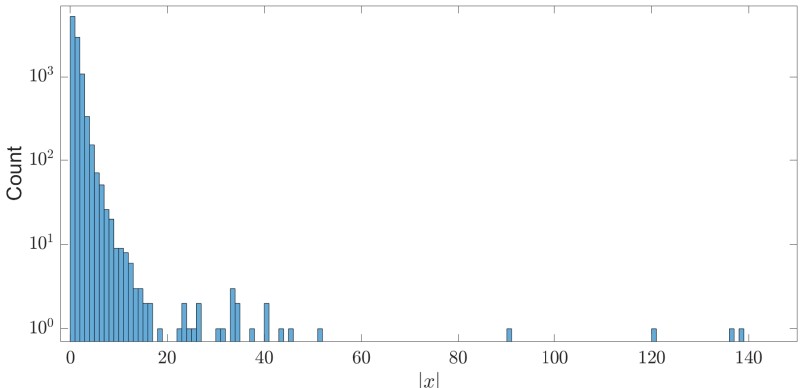

Figure S14: Histogram of $|x|$ for a sequence of i.i.d. random variables distributed according to $\mathcal{S}\alpha\mathcal{S}(1)$, where $\alpha = 1.7$.

$\{\varepsilon_l\}_{l=1}^L : \varepsilon_l > 0$, $\{\kappa_l\}_{l=1}^L : \kappa_l \in (0,1)$, and $p \geq \max_l \alpha_l$ there exists $\{d'_{l,0}\}_{l=1}^L : d'_{l,0} \in \mathbb{N}$ such that for $d_l \geq d'_{l,0}$, $l \in [\![1, L]\!]$,

$$\mathbb{P}\left[\left(\|\mathbf{x}_l^{(\kappa_l d_l)} - \mathbf{x}_l\|_p/\|\mathbf{x}_l\|_p\right) \leq \varepsilon_l\right] - \mathbb{P}\left[\left(\|\mathbf{w}_l^{(\kappa_l d_l)} - \mathbf{w}_l\|_p/\|\mathbf{w}_l\|_p\right) \leq \varepsilon_l\right] \leq \epsilon. \quad \text{(S11)}$$

Moreover due to Lemma S2, there exists $\{d''_{l,0}\}_{l=1}^L : d''_{l,0} \in \mathbb{N}$ such that for $d_l \geq d''_{l,0}$,

$$\mathbb{P}\left[\left(\|\mathbf{x}_l^{(\kappa_l d_l)} - \mathbf{x}_l\|_p/\|\mathbf{x}_l\|_p\right) \leq \varepsilon_l\right] = 1 \quad \text{(S12)}$$

The results follows from (S12) and (S11) and by choosing $d_{l,0} \geq \max(d'_{l,0}, d''_{l,0})$.

$\square$

### S5.3 Proof of Theorem 2

*Proof.* Fix $l \in [\![2, L-1]\!]$ and recall that $\mathbf{W}_l \in \mathbb{R}^{h_l \times h_{l-1}}$ with $d_l = h_l \times h_{l-1}$. Define

$$\mathbf{X}_l := \frac{1}{h_l^{2/\alpha_l}} \mathbf{W}_l^\top \mathbf{W}_l \quad \text{(S13)}$$

and denote the eigenvalues of $\mathbf{X}_l$ by $\boldsymbol{\lambda}_l = [\lambda_{l,1}, \dots, \lambda_{l,h_{l-1}}]$.

Let $\mathbf{U}_l \in \mathbb{R}^{h_l \times h_{l-1}}$ be a matrix whose entries are independent and identically distributed from a symmetric stable distribution with tail index $\alpha_l$. Note that $\mathbf{W}_l$ converges in distribution to $\mathbf{U}_l$, as network dimension goes to infinity, due to HML condition and the assumptions of the theorem. Similarly, define

$$\mathbf{X}'_l := \frac{1}{h_l^{2/\alpha_l}} \mathbf{U}_l^\top \mathbf{U}_l \quad \text{(S14)}$$

and denote the eigenvalues of $\mathbf{X}'_l$ by $\boldsymbol{\lambda}'_l = [\lambda'_{l,1}, \dots, \lambda'_{l,h_{l-1}}]$.

As $\mathbf{W}_l$ converges in distribution to $\mathbf{U}_l$, then $\lambda_{l,k}$ also weakly converges to $\lambda'_{l,k}$, due to Weyl's inequality ([Bha97, Page 63]). Hence, for any $\epsilon > 0$, $\{\varepsilon_l\}_{l=2}^{L-1} : \varepsilon_l > 0$, and $\{\kappa_l\}_{l=2}^{L-1} : \kappa_l \in (0,1)$, there exists $\{\hat{h}_{l,0}\}_{l=1}^{L-1}$, such that for every $l \in [\![2, L-1]\!]$, and $h_i \geq \hat{h}_{i,0}$ for all $i \in [\![1, l]\!]$, the following holds

$$\left| \mathbb{P}\left[\frac{\|\boldsymbol{\lambda}_i^{(\kappa_i h_{i-1})} - \boldsymbol{\lambda}_i\|_p}{\|\boldsymbol{\lambda}_i\|_p} \leq \varepsilon_i^2\right] - \mathbb{P}\left[\frac{\|\boldsymbol{\lambda}_i'^{(\kappa_i h_{i-1})} - \boldsymbol{\lambda}'_i\|_p}{\|\boldsymbol{\lambda}'_i\|_p} \leq \varepsilon_i^2\right] \right| \leq \frac{\epsilon}{2}. \quad \text{(S15)}$$

Moreover, since each $[\mathbf{U}]_{i,j}$ is independent and identically distributed from a symmetric stable distribution with tail index $\alpha_l$, by [TTR$^+$20, Theorem 2.7], as $h_l \to \infty$, for each $k = 1, \dots, h_{l-1}$,

the eigenvalue $\lambda'_{l,k}$ weakly converges to a random variable $\xi_{l,k}$, where the collection $\{\xi_{l,k}\}_{k=1}^{h_{l-1}}$ is independent and identically distributed from a positive stable distribution with tail index $\alpha_l/2$. Denote by $\boldsymbol{\xi}_l := [\xi_{l,1}, \ldots, \xi_{l,h_{l-1}}] \in \mathbb{R}^{h_{l-1}}$ the random vector containing the limiting i.i.d. random variables.

We will now construct a sequence of $\{h_{l,0}\}_{l=1}^{L-1}$ such that the claims will follow for $h_l \geq h_{l,0}$. Let us start from the second layer, i.e., set $l = 2$. Then, by Lemma S2, for any $\varepsilon_2 > 0$, $\kappa_2 \in (0, 1)$, and $p \geq \alpha_2/2$, there exists $h'_{1,0} \in \mathbb{N}_+$, such that $h_1 \geq h'_{1,0}$ implies:

$$\mathbb{P}\left[\frac{\|\boldsymbol{\xi}_2^{(\kappa_2 h_1)} - \boldsymbol{\xi}_2\|_p}{\|\boldsymbol{\xi}_2\|_p} \leq \varepsilon_2^2\right] = 1. \tag{S16}$$

Let $h_{1,0} = h'_{1,0} \vee \hat{h}_{1,0}$. Having fixed $h_1 \geq h_{1,0}$, we now iterate the following argument from $l = 2$ to $l = L - 1$ sequentially. Due to the weak convergence of the eigenvalues, we have:

$$\lim_{h_l \to \infty} \mathbb{P}\left[\frac{\|\boldsymbol{\lambda}_l^{'(\kappa_l h_{l-1})} - \boldsymbol{\lambda}'_l\|_p}{\|\boldsymbol{\lambda}'_l\|_p} \leq \varepsilon_l^2\right] = \mathbb{P}\left[\frac{\|\boldsymbol{\xi}_l^{(\kappa_l h_{l-1})} - \boldsymbol{\xi}_l\|_p}{\|\boldsymbol{\xi}_l\|_p} \leq \varepsilon_l^2\right] \tag{S17}$$

$$= 1. \tag{S18}$$

Hence, combining with (S15), for any $\epsilon > 0$, there exists $h''_{l,0} \in \mathbb{N}_+$, such that $h_l \geq h''_{l,0} \vee \hat{h}_{l,0}$ implies

$$\mathbb{P}\left[\frac{\|\boldsymbol{\lambda}_l^{(\kappa_l h_{l-1})} - \boldsymbol{\lambda}_l\|_p}{\|\boldsymbol{\lambda}_l\|_p} \leq \varepsilon_l^2\right] \geq 1 - \epsilon. \tag{S19}$$

If $l = L - 1$, set $h_{l,0} = h''_{l,0} \vee \hat{h}_l \vee h_{l-1}$. If $l \leq L - 2$, repeat the previous argument to find a $h'_{l,0}$, such that $h_l \geq h'_{l,0}$ implies

$$\mathbb{P}\left[\frac{\|\boldsymbol{\xi}_{l+1}^{(\kappa_{l+1} h_l)} - \boldsymbol{\xi}_{l+1}\|_p}{\|\boldsymbol{\xi}_{l+1}\|_p} \leq \varepsilon_{l+1}^2\right] = 1, \tag{S20}$$

and set $h_{l,0} = h'_{l,0} \vee h''_{l,0} \vee \hat{h}_{l,0} \vee h_{l-1}$. This proves the first claim.

To prove the second claim, first notice that we can set $p = 1$ as $\max_l \alpha_l < 2$, hence $p = 1 \geq \alpha_l/2$ for all $l$. Now, fix $l \in [\![2, L-1]\!]$, and for a given $h_{l-1}$ and $h_l$, consider the singular value decomposition of $\mathbf{W}_l$ as:

$$\mathbf{W}_l = \mathbf{U}\boldsymbol{\Sigma}\mathbf{V}^\top, \tag{S21}$$

and define $\mathbf{W}_l^{[\kappa_l h_l - 1]} := \mathbf{U}\boldsymbol{\Sigma}^{(\kappa_l h_{l-1})}\mathbf{V}^\top$, where $\boldsymbol{\Sigma}^{(\kappa_l h_{l-1})}$ is the diagonal matrix whose diagonal entries contain the $\lceil \kappa_l h_{l-1} \rceil$ largest singular values (i.e., prune the diagonal part of $\boldsymbol{\Sigma}$). By using (S19) and the fact that the Schatten 2-norm coincides with the Frobenius norm, we have:

$$\|\mathbf{W}_l\|^2 = h_l^{1/\alpha_l}\|\boldsymbol{\lambda}_l\|_1, \quad \text{and} \quad \|\mathbf{W}_l^{[\kappa_l h_l - 1]} - \mathbf{W}_l\|^2 = h_l^{1/\alpha_l}\|\boldsymbol{\lambda}_l^{(\kappa_l h_{l-1})} - \boldsymbol{\lambda}_l\|_1. \tag{S22}$$

Hence, we conclude that for $h_l \geq h_{l,0}$, the following inequality holds for $l \in [\![2, L-1]\!]$:

$$\mathbb{P}\left[\frac{\|\mathbf{W}_l^{[\kappa_l h_l - 1]} - \mathbf{W}_l\|}{\|\mathbf{W}_l\|} \leq \varepsilon_l\right] \geq 1 - \epsilon. \tag{S23}$$

This concludes the proof. $\qquad\square$

## S5.4 Proof of Theorem 3

*Proof.* For $l \in [\![2, L]\!]$, let $v_{l,i} = \|\mathbf{W}_l(i)\|_p$, where $\mathbf{W}_l(i) \in \mathbb{R}^{h_l}$ is the $i$-th column of $\mathbf{W}_l \in \mathbb{R}^{h_l \times h_{l-1}}$ for $i \in [\![1, h_{l-1}]\!]$. Note that by definition, for any $\kappa_l \in (0, 1)$

$$\|\mathbf{v}_l\|_p = \|\mathbf{w}_l\|_p, \quad \text{and} \quad \|\mathbf{v}_l^{(\kappa_l h_{l-1})} - \mathbf{v}_l\|_p = \|\mathbf{w}_l^{\{\kappa_l h_{l-1}\},p} - \mathbf{w}_l\|_p.$$

Hence, it suffices to show that for any $\epsilon > 0$ and $\varepsilon_l > 0$, there exists $h_{l-1,0}$ such that for $h_{l-1} \geq h_{l-1,0}$,

$$\mathbb{P}\left[\left(\|\mathbf{v}_l^{(\kappa_l h_{l-1})} - \mathbf{v}_l\|_p\right)/\|\mathbf{v}_l\|_p \leq \varepsilon_l\right] \geq 1 - \epsilon. \tag{S24}$$

As network dimensions grow, due to HML condition, $w_{l,j}, j \in [\![1, d_l]\!]$ converges in distribution to an i.i.d. heavy-tailed random variable with tail index $\alpha_l$. Hence, as $\{h_{l-1}\}_{l=2}^{L}$ grows, $v_{l,i}$ also converges in distribution to a heavy-tailed random variable, denoted as $\mathbf{x}_l = (x_{l,1}, \ldots, x_{l,h_{l-1}}) \in \mathbb{R}^{h_{l-1}}$ , with i.i.d. elements and tail index $\alpha_l \in (0, 2)$. Thus, there exists $\{h'_{l-1,0}\}_{l=2}^{L}: h'_{l-1,0} \in \mathbb{N}$ such that for $h_{l-1} \geq h'_{l-1,0}, l \in [\![2, L]\!]$,

$$\mathbb{P}\left[\left(\|\mathbf{x}_l^{(\kappa_l h_{l-1})} - \mathbf{x}_l\|_p/\|\mathbf{x}_l\|_p\right) \leq \varepsilon_l\right] - \mathbb{P}\left[\left(\|\mathbf{v}_l^{(\kappa_l h_{l-1})} - \mathbf{v}_l\|_p/\|\mathbf{v}_l\|_p\right) \leq \varepsilon_l\right] \leq \epsilon. \tag{S25}$$

Moreover due to Lemma S2, there exists $\{h''_{l-1,0}\}_{l=2}^{L}: h''_{l-1,0} \in \mathbb{N}$ such that for $h_{l-1} \geq h''_{l-1,0}$,

$$\mathbb{P}\left[\left(\|\mathbf{x}_l^{(\kappa_l h_{l-1})} - \mathbf{x}_l\|_p/\|\mathbf{x}_l\|_p\right) \leq \varepsilon_l\right] = 1 \tag{S26}$$

The results follows from (S12) and (S11) and by choosing $h_{l-1,0} \geq \max(h'_{l-1,0}, h''_{l-1,0})$. $\qquad\square$

### S5.5 Proof of Theorem 4

Let

$$D(y, f_\mathbf{w}(\mathbf{x})) := f_\mathbf{w}(\mathbf{x})[y] - \max_{j \neq y} f_\mathbf{w}(\mathbf{x})[j].$$

Define the surrogate loss function $\ell_{\gamma,\tau}: \mathcal{Y} \times \mathcal{Y} \mapsto [0, 1]$, with margin loss $\gamma \geq 0$ and continuity margin $\tau > 0$, for the multiclass classifier $f_\mathbf{w}$ as:

$$\ell_{\gamma,\tau}(y, f_\mathbf{w}(\mathbf{x})) := \begin{cases} 1, & \text{if } D(y, f_\mathbf{w}(\mathbf{x})) \leq \gamma, \\ 1 - \frac{D(y, f_\mathbf{w}(\mathbf{x})) - \gamma}{\tau}, & \text{if } \gamma < D(y, f_\mathbf{w}(\mathbf{x})) \leq \gamma + \tau, \\ 0, & \text{if } \gamma + \tau < D(y, f_\mathbf{w}(\mathbf{x})). \end{cases} \tag{S27}$$

Note that $\ell_\gamma(y, f_\mathbf{w}(\mathbf{x})) = \ell_{\gamma,0}(y, f_\mathbf{w}(\mathbf{x}))$. Population and empirical risks of a hypothesis $\mathbf{w}$ are denoted by $\mathcal{R}_{\gamma,\tau}(\mathbf{w})$ and $\hat{\mathcal{R}}_{\gamma,\tau}(\mathbf{w})$, respectively.

*Proof.* Recall that for all $l \in [\![1, L]\!]$, $\widehat{\mathbf{w}}_l := \mathbf{w}_l^{(\kappa_l d_l)}$. Denote by $A$ the event that $\mathbf{w}$ is compressible, *i.e.* when for all $l \in [\![1, L]\!]$, $\|\widehat{\mathbf{w}}_l - \mathbf{w}_l\| \leq \varepsilon\|\mathbf{w}_l\|$. Denote its complement by $A^C$ and note that $\mathbb{P}(A^C) \leq \epsilon$, where the probability is with respect to $P_{S,\mathbf{w}}$.

Fix $\delta, \tau > 0$. We show that with probability of at least $1 - 2e^{-\kappa d/2} - \delta - \epsilon$:

$$\mathcal{R}_{0,\tau}(\widehat{\mathbf{w}}) \leq \hat{\mathcal{R}}_{0,\tau}(\widehat{\mathbf{w}}) + \left(12\mathcal{L}(\tau, \delta)R(\delta) + \sqrt{d}\right)\sqrt{\frac{(\kappa + \epsilon_\kappa)d\log(n)}{n}}, \tag{S28}$$

and moreover $\|\mathbf{w}\| \leq R(\delta)$ and $\|\widehat{\mathbf{w}}_l - \mathbf{w}_l\| \leq \varepsilon\|\mathbf{w}_l\|$ for all $l \in [\![1, L]\!]$, simultaneously. Then, under the latter two conditions,

$$\mathcal{R}_0(\widehat{\mathbf{w}}) \leq \mathcal{R}_{0,\tau}(\widehat{\mathbf{w}}),$$
$$\hat{\mathcal{R}}_{0,\tau}(\widehat{\mathbf{w}}) \leq \hat{\mathcal{R}}_{\gamma(\delta,\tau)}(\mathbf{w}),$$

using Lemma S1, Definition S27, and Lemma S3 that bounds the Lipschitz coefficient of the network. This completes the proof.

**Lemma S3.** *Suppose that for $l \in [\![1, L]\!]$ and a given $\upsilon_l > 0$, we have $\|\mathbf{w}_l - \mathbf{w}'_l\| \leq \upsilon_l$.*

    *i. Then, for any $(\mathbf{x}, y)$, the following relations hold:*

$$\|f_\mathbf{w}(\mathbf{x}) - f_{\mathbf{w}'}(\mathbf{x})\| \leq B\prod_{l=1}^{L}(\|\mathbf{w}_l\| + \upsilon_l) - B\prod_{l=1}^{L}\|\mathbf{w}_l\|,$$

$$\left|\ell_{0,\tau}(y, f_\mathbf{w}(\mathbf{x})) - \ell_{0,\tau}(y, f_{\mathbf{w}'}(\mathbf{x}))\right| \leq \frac{\sqrt{2}}{\tau}\|f_\mathbf{w}(\mathbf{x}) - f_{\mathbf{w}'}(\mathbf{x})\|.$$

*ii. In particular, if $v_l = v \leq R/\sqrt{L}$ for $l \in [\![1, L]\!]$ and if $\|\mathbf{w}\| \leq R$, then*

$$\|f_{\mathbf{w}}(\mathbf{x}) - f_{\mathbf{w}'}(\mathbf{x})\| \leq BL\left(\frac{2R}{\sqrt{L}}\right)^{L-1} v,$$

$$\left|\ell_{0,\tau}(y, f_{\mathbf{w}}(\mathbf{x})) - \ell_{0,\tau}(y, f_{\mathbf{w}'}(\mathbf{x}))\right| \leq \frac{BL\sqrt{2}}{\tau}\left(\frac{2R}{\sqrt{L}}\right)^{L-1} v =: \mathcal{L}(\tau, \delta)v.$$

Hence, it remains to show (S28) together with the conditions $\|\mathbf{w}\| \leq R(\delta)$ and $\|\widehat{\mathbf{w}}_l - \mathbf{w}_l\| \leq \varepsilon\|\mathbf{w}_l\|$, for $l \in [\![1, L]\!]$, hold with probability at least $1 - 2e^{-\kappa d} - \delta - \epsilon$. Now, first whenever $\|\widehat{\mathbf{w}}\| \leq R(\delta)$, we discretize $\widehat{\mathbf{w}}$. Let

$$\widehat{\mathcal{W}}(R(\delta), d, \kappa) := \left\{\widehat{\mathbf{w}} \in \mathbb{R}^d \big| \|\widehat{\mathbf{w}}\| \leq R(\delta), \|\widehat{\mathbf{w}}\|_0 \leq \kappa d\right\},$$

where $\|\widehat{\mathbf{w}}\|_0$ denotes the number of non-zero components of $\widehat{\mathbf{w}}$. Assume that $\widetilde{\mathcal{W}}(R(\delta), d, \kappa)$ is a discretization of this space with $v > 0$ precision, *i.e.* for every $\widehat{\mathbf{w}} \in \widehat{\mathcal{W}}(R(\delta), d, \kappa)$, there exists $\widetilde{\mathbf{w}} \in \widetilde{\mathcal{W}}(R(\delta), d, \kappa)$ satisfying $\|\widetilde{\mathbf{w}} - \widehat{\mathbf{w}}\| \leq v$. Among all such coverings, consider the one with minimum number of $\mathcal{N}_v$ points.

**Lemma S4.** $\mathcal{N}_v \leq e^{dh_b(\kappa)}\left(\frac{3R(\delta)}{v}\right)^{\kappa d}$.

Note that in general $\|\widehat{\mathbf{w}}\| \leq \|\mathbf{w}\|$. Let $l := c_1(\delta, \tau)\sqrt{\frac{(\kappa+\epsilon_\kappa)d\log(n)}{n}}$ and $v = \frac{al}{4\mathcal{L}(\tau,\delta)}$, where $a := \frac{12\mathcal{L}(\tau,\delta)R(\delta)}{12\mathcal{L}(\tau,\delta)R(\delta)+\sqrt{d}}$. Then

$$\mathbb{P}\left(\left|\mathcal{R}_{0,\tau}(\widehat{\mathbf{w}}) - \hat{\mathcal{R}}_{0,\tau}(\widehat{\mathbf{w}})\right| \geq l \bigcup \|\mathbf{w}\| > R(\delta) \bigcup A^C\right)$$

$$\leq \mathbb{P}\left(\left|\mathcal{R}_{0,\tau}(\widehat{\mathbf{w}}) - \hat{\mathcal{R}}_{0,\tau}(\widehat{\mathbf{w}})\right| \geq l \bigcap \|\mathbf{w}\| \leq R(\delta)\right) + \mathbb{P}(\|\mathbf{w}\| \geq R(\delta)) + \mathbb{P}\left(A^C\right)$$

$$\leq \mathbb{P}\left(\left|\mathcal{R}_{0,\tau}(\widehat{\mathbf{w}}) - \hat{\mathcal{R}}_{0,\tau}(\widehat{\mathbf{w}})\right| \geq l \bigcap \|\mathbf{w}\| \leq R(\delta)\right) + \delta + \epsilon$$

$$\leq \mathbb{P}\left(\sup_{\widehat{\mathbf{w}} \in \widehat{\mathcal{W}}(R(\delta), d, \kappa)} \left|\mathcal{R}_{0,\tau}(\widehat{\mathbf{w}}) - \hat{\mathcal{R}}_{0,\tau}(\widehat{\mathbf{w}})\right| \geq l/2\right) + \delta + \epsilon$$

$$\overset{(a)}{\leq} \mathbb{P}\left(\max_{\widetilde{\mathbf{w}} \in \widetilde{\mathcal{W}}(R(\delta), d, \kappa)} \left|\mathcal{R}_{0,\tau}(\widetilde{\mathbf{w}}) - \hat{\mathcal{R}}_{0,\tau}(\widetilde{\mathbf{w}})\right| \geq l(1-a)/2\right) + \delta + \epsilon$$

$$\overset{(b)}{\leq} \mathcal{N}_v \max_{\widetilde{\mathbf{w}} \in \widetilde{\mathcal{W}}(R(\delta), d, \kappa)} \mathbb{P}\left(\left|\mathcal{R}_{0,\tau}(\widetilde{\mathbf{w}}) - \hat{\mathcal{R}}_{0,\tau}(\widetilde{\mathbf{w}})\right| \geq l(1-a)/2\right) + \delta + \epsilon$$

$$\overset{(c)}{\leq} 2\mathcal{N}_v \exp\left(-nl^2(1-a)^2/2\right) + \delta + \epsilon$$

$$\overset{(d)}{\leq} 2\exp\left(-nl^2(1-a)^2/2 + \kappa d\log\left(\frac{12\mathcal{L}(\tau,\delta)R(\delta)}{al}\right) + dh_b(\kappa)\right) + \delta + \epsilon, \quad \text{(S29)}$$

where $(a)$ is derived since

$$\left|\mathcal{R}_{0,\tau}(\widehat{\mathbf{w}}) - \hat{\mathcal{R}}_{0,\tau}(\widehat{\mathbf{w}})\right| \leq \left|\mathcal{R}_{0,\tau}(\widetilde{\mathbf{w}}) - \hat{\mathcal{R}}_{0,\tau}(\widetilde{\mathbf{w}})\right| + 2\mathcal{L}(\tau, \delta)v = \left|\mathcal{R}_{0,\tau}(\widetilde{\mathbf{w}}) - \hat{\mathcal{R}}_{0,\tau}(\widetilde{\mathbf{w}})\right| + al/2.$$

by Lemma S3, given that $v \leq R/\sqrt{L}$, and the triangle inequality. The inequality $(b)$ is obtained by applying the union bound, $(c)$ is derived using Hoeffding's inequality and since loss is bounded by 1, and $(d)$ is due to Lemma S4.

It remains to show that the term in the exponent in (S29) is upper bounded by $-\kappa d/2$ and $\upsilon \le R/\sqrt{L}$.

$$-nl^2(1-a)^2/2 + \kappa d \log\left(\frac{12\mathcal{L}(\tau,\delta)R(\delta)}{al}\right) + dh_b(\kappa) \tag{S30}$$

$$= -\frac{c_1(\delta,\tau)^2(1-a)^2\kappa d\log(n)}{2} + \frac{\kappa d}{2}\log(n) \tag{S31}$$

$$+ \kappa d\log\left(\frac{12\mathcal{L}(\tau,\delta)R(\delta)}{ac_1(\delta,\tau)\sqrt{d}}\right) \tag{S32}$$

$$- \frac{c_1(\delta,\tau)^2(1-a)^2\epsilon_\kappa d\log(n)}{2} - \frac{\kappa d}{2}\log(\kappa+\epsilon_\kappa) + dh_b(\kappa) \tag{S33}$$

$$- \frac{\kappa d}{2}\log\log(n). \tag{S34}$$

It can be verified that (S31) and (S32) are non-positive when $c_1(\delta,\tau) \ge \left(12\mathcal{L}(\tau,\delta)R(\delta) + \sqrt{d}\right)/\sqrt{d}$.
Moreover, with this choice of $c_1(\delta,\tau)$, (S33) is non-positive for $\epsilon_\kappa = (2h_b(\kappa) - \kappa\log(\kappa))/(\log(n))$.
Finally, (S33) is less than $-\kappa d/2$, for $n \ge 16$.

Finally with the chosen value of $\upsilon$, $\upsilon \le R/\sqrt{L}$ holds if $n/\log(n) \ge 10L$. This completes the proof. $\qquad\square$

### S5.6 Proof of Corollary 1

For notation convenience, let $\mathcal{S}\alpha\mathcal{S}_n(\sigma) \equiv \mathcal{S}\alpha\mathcal{S}(\sigma_\alpha\sigma)$, where $n$ stands for normalized and $\sigma_\alpha := (2\Gamma(-\alpha)\cos((2-\alpha)\pi/2))^{1/\alpha}$. First, we state the Corollary for a more general case, and then we state the proof of this general result.

**Corollary S2.** *Assume that for $l \in [\![1,L]\!]$ and $i \in [\![1,d_l]\!]$, the conditional distribution of $w_{l,i} \overset{i.i.d.}{\sim} \mathcal{S}\alpha_l\mathcal{S}_n(\sigma_l)$ with $\alpha_l \in (1,2)$. Further assume that $\sigma^2 := \sum_{l=1}^L (d_l/d)\sigma_l^2$ and $\{\alpha_l\}_{l=1}^L$ do not depend on $S$. Then for every $\varepsilon > 0$, $\kappa_l \in (0,1)$, $l \in [\![1,L]\!]$, and $\beta > 0$, there exists $d_{l,0} \in \mathbb{N}$, such that for every $n\colon n/\log(n) \ge 10L$ and $\tau > 0$, with probability at least $1 - 3d^{-\beta}$,*

$$\mathcal{R}_0(\widehat{\mathbf{w}}) \le \hat{\mathcal{R}}_\gamma(\mathbf{w}) + \left(a(\alpha)\sigma^L d^{\frac{L(\alpha+2\beta+2)}{2\alpha}}/\tau + \sqrt{d}\right)\sqrt{(\kappa+\epsilon_\kappa)\log(n)/n}, \tag{S35}$$

*where $\{\widehat{\mathbf{w}}_1\}_l = \{\mathbf{w}_l^{(\kappa d_l)}\}_l$, $\alpha := \min_l \alpha_l$, $a(\alpha) := 6\sqrt{2}B6^L4^{L/\alpha}/L^{(L-3)/2}$, $\gamma := \tau + b_\varepsilon(\alpha)\sigma^L d^{\frac{L(\alpha+2\beta+2)}{2\alpha}}/\tau$, and $b_\varepsilon(\alpha) := \sqrt{2}B3^L4^{L/\alpha}\left((1+\varepsilon)^L - 1\right)/L^{L/2}$.*

*Proof.* First, given any $S$, we bound the term $R_S(\delta)$, defined as

$$R_S(\delta) := \inf\left\{R\colon \mathbb{P}\left(\|\mathbf{w}\| \ge R|S\right) \le \delta\right\}. \tag{S36}$$

**Lemma S5.** *If for $l \in [\![1,L]\!]$, $\mathbf{x}_l$ is an i.i.d. $d_l$-dimensional vector with with $\mathcal{S}\alpha_l\mathcal{S}_n(\sigma_l)$ distributions and $\alpha_l \in (1,2)$, then for $\delta < 2d(2 - \max_l \alpha_l)^\alpha$*

$$\inf\{R\colon \mathbb{P}(\|\mathbf{x}\| \ge R) \le \delta\} \le 3\sigma\sqrt{d}\left(\frac{4d}{\delta}\right)^{1/\alpha},$$

*where $\sigma := \sqrt{\sum_{l=1}^L (d_l/d)\sigma_l^2}$ and $\alpha := \min_l \alpha_l$.*

Hence, $R_S(\delta) \le 3\sigma\sqrt{d}\left(\frac{4d}{\delta}\right)^{1/\alpha}$. Since $\sigma$, $\alpha$, and $\max_l \alpha_l$ do not depend on $S$, then this bound is the same for all $S$. Thus, for $\delta < 2d(2 - \max_l \alpha_l)^\alpha$,

$$R(\delta) \le 3\sigma\sqrt{d}\left(\frac{4d}{\delta}\right)^{1/\alpha}. \tag{S37}$$

Next, due to Lemma S2, the assumption **H** 1 holds for any $\varepsilon > 0$ and $\{\kappa_l\}_{l=1}^L\colon \kappa_l \in (0,1)$ and some $\{d_{l,0}\}_{l=1}^L\colon d_{l,0} \in \mathbb{N}$, with $\epsilon = 0$. Note that $d_{l,0}$ does not depend on $S$ as $\{\alpha_l\}_{l=1}^L$ is independent of $S$. The proof follows now by Theorem 4 with $\delta = d^{-\beta}$ and using the relation (S37) when $2d_0^{\beta+1}(2 - \max_l \alpha_l)^\alpha \ge 1$ and $d_0/\log(d_0) \ge \beta/\kappa$. $\qquad\square$

## S5.7   Proof of Theorem S6

*Proof.* The proof of this theorem is similar to the proof of Theorem 4. Fix $\delta, \tau > 0$. We show that with probability of at least $1 - 2e^{-d/2} - \delta - \epsilon$:

$$\mathcal{R}_{0,\tau}(\mathbf{w}) \leq \hat{\mathcal{R}}_{0,\tau}(\mathbf{w}) + \max\left(2, 24\rho_\varepsilon(\kappa, d)\mathcal{L}(\tau, \delta)R(\delta)/\sqrt{d}\right)\sqrt{d\log(n)/n}, \qquad \text{(S38)}$$

where $\mathcal{R}_{0,\tau}(\cdot)$ and $\hat{\mathcal{R}}_{0,\tau}(\cdot)$ are defined in (S27). The claim follows then by noting that

$$\mathcal{R}_0(\mathbf{w}) \leq \mathcal{R}_{0,\tau}(\mathbf{w}),$$
$$\hat{\mathcal{R}}_{0,\tau}(\mathbf{w}) \leq \hat{\mathcal{R}}_\tau(\mathbf{w}),$$

due to (S27).

Recall that for all $l \in [\![1, L]\!]$, $\widehat{\mathbf{w}}_l := \mathbf{w}_l^{(\kappa_l d_l)}$. Denote by $A$ the event that $\mathbf{w}$ is compressible, *i.e.* when for all $l \in [\![1, L]\!]$, $\|\widehat{\mathbf{w}}_l - \mathbf{w}_l\| \leq \varepsilon\|\mathbf{w}_l\|$. Denote its complement by $A^C$ and note that $\mathbb{P}(A^C) \leq \epsilon$, where the probability is with respect to $P_\mathbf{w}$.

In the following, first we discretize $\mathbf{w}$ whenever $\|\mathbf{w}\| \leq R(\delta)$. Let

$$\mathcal{W}(R(\delta), \varepsilon, d, \kappa) := \left\{\mathbf{w} \in \mathbb{R}^d \big| \|\mathbf{w}\| \leq R(\delta), \|\mathbf{w}^{(\kappa d)} - \mathbf{w}\| \leq \varepsilon R(\delta)\right\}.$$

Assume that $\mathcal{W}'(R(\delta), \varepsilon, d, \kappa)$ is the discretization of this space with $\upsilon > 0$ precision, *i.e.* for every $\mathbf{w} \in \mathcal{W}(R(\delta), \varepsilon, d, \kappa)$, there exists $\mathbf{w}' \in \mathcal{W}'(R(\delta), \varepsilon, d, \kappa)$ satisfying $\|\mathbf{w}' - \mathbf{w}\| \leq \upsilon$. Among all such coverings, consider the one with minimum number of $\mathcal{N}'_\upsilon$ points.

**Lemma S6.** *For $d \geq 10$, if $\upsilon < \varepsilon R(\delta)$, then $\mathcal{N}'_\upsilon \leq \left(\frac{3\rho_\varepsilon(\kappa, d)R}{\upsilon}\right)^d$.*

Similar to the proof of Theorem 4 and by letting $l := c_2(\delta, \tau, \kappa)\sqrt{\frac{d\log(n)}{n}}$ and $\upsilon = \frac{al}{4\mathcal{L}(\tau, \delta)}$, where $a := \frac{12\rho_\varepsilon(\kappa, d)\mathcal{L}(\tau, \delta)R(\delta)}{12\rho_\varepsilon(\kappa, d)\mathcal{L}(\tau, \delta)R(\delta) + \sqrt{d}}$,

$$\mathbb{P}\left(\left|\mathcal{R}_{0,\tau}(\mathbf{w}) - \hat{\mathcal{R}}_{0,\tau}(\mathbf{w})\right| \geq l\right)$$

$$\leq \mathbb{P}\left(\left|\mathcal{R}_{0,\tau}(\mathbf{w}) - \hat{\mathcal{R}}_{0,\tau}(\mathbf{w})\right| \geq l \bigcup \|\mathbf{w}\| > R(\delta) \bigcup A^C\right)$$

$$\leq \mathbb{P}\left(\left|\mathcal{R}_{0,\tau}(\mathbf{w}) - \hat{\mathcal{R}}_{0,\tau}(\mathbf{w})\right| \geq l \bigcap \|\mathbf{w}\| \leq R(\delta) \bigcap A\right) + \delta + \epsilon$$

$$\leq \mathbb{P}\left(\sup_{\mathbf{w} \in \mathcal{W}(R(\delta), \varepsilon, d, \kappa)} \left|\mathcal{R}_{0,\tau}(\mathbf{w}) - \hat{\mathcal{R}}_{0,\tau}(\mathbf{w})\right| \geq l/2\right) + \delta + \epsilon$$

$$\overset{(a)}{\leq} \mathcal{N}'_\upsilon \max_{\mathbf{w}' \in \mathcal{W}'(R(\delta), \varepsilon, d, \kappa)} \mathbb{P}\left(\left|\mathcal{R}_{0,\tau}(\mathbf{w}') - \hat{\mathcal{R}}_{0,\tau}(\mathbf{w}')\right| \geq l(1 - a)/2\right) + \delta + \epsilon$$

$$\overset{(b)}{\leq} 2\exp\left(-\frac{nl^2(1 - a)^2}{2} + d\log\left(\frac{12\rho_\varepsilon(\kappa, d)\mathcal{L}(\tau, \delta)R(\delta)}{al}\right)\right) + \delta + \epsilon, \qquad \text{(S39)}$$

where $(a)$ holds when $\upsilon \leq R/\sqrt{L}$ and $(b)$ holds using Lemma S6 if $\upsilon < \varepsilon R(\delta)$.

It remains to show that the term in the exponent in (S39) is upper bounded by $-d/2$, $\upsilon < \varepsilon R(\delta)$, and $\upsilon \leq R/\sqrt{L}$. To show the first claim, we can write

$$-\frac{nl^2(1 - a)^2}{2} + d\log\left(\frac{12\rho_\varepsilon(\kappa, d)\mathcal{L}(\tau, \delta)R(\delta)}{al}\right) = -\frac{c_2(\delta, \tau, \kappa)^2(1 - a)^2 d\log(n)}{2} + \frac{d}{2}\log(n)$$

$$\text{(S40)}$$

$$+ d\log\left(\frac{12\rho_\varepsilon(\kappa, d)\mathcal{L}(\tau, \delta)R(\delta)}{ac_2(\delta, \tau, \kappa)\sqrt{d}}\right) \qquad \text{(S41)}$$

$$- \frac{d}{2}\log\log(n). \qquad \text{(S42)}$$

It can be verified that (S40) and (S41) are non-positive when $c_2(\delta, \tau, \kappa) = \left(12\rho_\varepsilon(\kappa, d)\mathcal{L}(\tau, \delta)R(\delta) + \sqrt{d}\right)/\sqrt{d}$ and (S42) is less than $-d/2$, for $n \geq 16$.

To verify $\upsilon < \varepsilon R(\delta)$, where $\upsilon = \frac{al}{4\mathcal{L}(\tau, \delta)}$, we have

$$
\begin{aligned}
\upsilon &= \frac{12\rho_\varepsilon(\kappa, d)\mathcal{L}(\tau, \delta)R(\delta)}{12\rho_\varepsilon(\kappa, d)\mathcal{L}(\tau, \delta)R(\delta) + \sqrt{d}} \times \frac{12\rho_\varepsilon(\kappa, d)\mathcal{L}(\tau, \delta)R(\delta) + \sqrt{d}}{\sqrt{d}} \times \frac{\sqrt{d\log(n)/n}}{4\mathcal{L}(\tau, \delta)} \\
&= 3\rho_\varepsilon(\kappa, d)R(\delta)\sqrt{\log(n)/n} \\
&\leq 9\varepsilon^{1-\kappa}R(\delta)\sqrt{\log(n)/n} \overset{(a)}{\leq} \varepsilon R(\delta),
\end{aligned}
$$

where $(a)$ holds when $\varepsilon^\kappa \geq 9\sqrt{\log(n)/n}$.

Moreover, with the chosen value of $\upsilon$, $\upsilon \leq R/\sqrt{L}$ holds if $n/\log(n) \geq 9L$. Finally note that $\left(12\rho_\varepsilon(\kappa, d)\mathcal{L}(\tau, \delta)R(\delta) + \sqrt{d}\right) \leq \max(24\rho_\varepsilon(\kappa, d)\mathcal{L}(\tau, \delta)R(\delta), 2\sqrt{d})$. This completes the proof. $\qquad \square$

## S5.8 Proof of Proposition S1

*Proof.* For ease of notations, for $\kappa, \varepsilon > 0$ and $\mathbf{w} \in \mathbb{R}^d$, let

$$\varepsilon_p(\mathbf{w}, \kappa) := \|\mathbf{w}^{(\kappa d)} - \mathbf{w}\|_p / \|\mathbf{w}\|_p \quad \text{and hence} \quad \kappa_p(\mathbf{w}, \varepsilon) := \min\{\kappa \colon \varepsilon_p(\mathbf{w}, \kappa) \leq \varepsilon\}. \tag{S43}$$

Let $\mathbf{x} = (|w_1|, \ldots, |w_d|)$ and let $x_{d,i}$ be the corresponding ordered sequence, *i.e.*

$$x_{d,1} \geq x_{d,2} \geq \cdots \geq x_{d,d}.$$

Let

$$\mathbf{y}^d = \frac{1}{a_d}(x_{d,1}, x_{d,2}, \ldots, x_{d,d}, 0, 0, \ldots) \in \mathbb{R}^\infty,$$

where $a_d$ is a normalizing constant defined in [LWZ81, Equation 3]. Moreover let $e_i$, $i = 1, 2, \ldots$, be i.i.d. standard exponential random variables with partial sum $\Gamma_i := \sum_{l=1}^i e_l$ and let $z_i(\alpha) := \Gamma_i^{-1/\alpha}$. Then, due to [LWZ81, Lemma 1],

$$\lim_{d \to \infty} \mathbf{y}^d \overset{\mathrm{d}}{=} (z_1(\alpha), z_2(\alpha), \ldots).$$

where $\mathrm{d}$ denotes convergence in distribution.

First, we show that for any $\kappa > 0$, $\varepsilon_p\big(\mathbf{z}^d(\alpha), \kappa\big)$ is increasing with respec to $\alpha$. This term can be written as

$$\varepsilon_p\big(\mathbf{z}^d(\alpha), \kappa\big)^p = \frac{\displaystyle\sum_{l=\lceil \kappa d \rceil + 1}^d z_l(\alpha)^p}{\displaystyle\sum_{l=1}^d z_l(\alpha)^p} = \frac{\displaystyle\sum_{l=\lceil \kappa d \rceil + 1}^d \Gamma_l^{-\frac{p}{\alpha}}}{\displaystyle\sum_{l=1}^d \Gamma_l^{-\frac{p}{\alpha}}} =: \frac{u}{v}.$$

Taking the derivative with respect to $\alpha$ gives

$$\frac{\partial \varepsilon_p\big(\mathbf{z}^d(\alpha), \kappa\big)^p}{\partial \alpha} = \frac{vu' - v'u}{v^2},$$

where

$$
\begin{aligned}
vu' - v'u &= \frac{p}{\alpha^2}\left[\left(\sum_{l=1}^d \Gamma_l^{-\frac{p}{\alpha}}\right)\left(\sum_{l=\lceil \kappa d \rceil + 1}^d \Gamma_l^{-\frac{p}{\alpha}}\log(\Gamma_l)\right) - \left(\sum_{l=1}^d \Gamma_l^{-\frac{p}{\alpha}}\log(\Gamma_l)\right)\left(\sum_{l=\lceil \kappa d \rceil + 1}^d \Gamma_l^{-\frac{p}{\alpha}}\right)\right] \\
&= \frac{p}{\alpha^2}\sum_{l_1=1}^{\lceil \kappa d \rceil}\sum_{l_2=\lceil \kappa d \rceil + 1}^d \Gamma_{l_1}^{-\frac{p}{\alpha}}\Gamma_{l_2}^{-\frac{p}{\alpha}}\left(\log(\Gamma_{l_2}) - \log(\Gamma_{l_1})\right) \\
&\overset{a.s.}{>} 0.
\end{aligned}
$$

This shows that $\varepsilon_p\big(\mathbf{z}^d(\alpha),\kappa\big)$ is almost surely strictly increasing with respect to $\alpha$, and consequently $\kappa_p\big(\mathbf{z}^d(\alpha),\varepsilon\big)$ is almost surely increasing with respect to $\alpha$.

Since $\varepsilon_p(\mathbf{w},\kappa)$ is a bounded function and almost surely continuous with respect to $\mathbf{w}$, $\mathbb{E}\big[\varepsilon_p\big(\mathbf{w}_i^d,\kappa\big)\big]$ converges also to $\mathbb{E}[\varepsilon_p(\mathbf{z}_i^\infty(\alpha_i),\kappa)]$, for $i=1,2$. To show (S6), choose $d_0(\delta)$ large enough, such that

$$\left|\mathbb{E}\big[\varepsilon_p\big(\mathbf{w}_1^d,\kappa\big)\big] - \mathbb{E}[\varepsilon_p(\mathbf{z}_1^\infty(\alpha_1),\kappa)]\right| < \frac{\delta}{4}, \quad \left|\mathbb{E}\big[\varepsilon_p\big(\mathbf{w}_2^d,\kappa\big)\big] - \mathbb{E}[\varepsilon_p(\mathbf{z}_2^\infty(\alpha_2),\kappa)]\right| < \frac{\delta}{4},$$

$$\left|\mathbb{E}\big[\varepsilon_p\big(\mathbf{z}_1^d,\kappa\big)\big] - \mathbb{E}[\varepsilon_p(\mathbf{z}_1^\infty(\alpha_1),\kappa)]\right| < \frac{\delta}{4}, \quad \left|\mathbb{E}\big[\varepsilon_p\big(\mathbf{z}_2^d,\kappa\big)\big] - \mathbb{E}[\varepsilon_p(\mathbf{z}_2^\infty(\alpha_2),\kappa)]\right| \le \frac{\delta}{4}.$$

Then,

$$\mathbb{E}\big[\varepsilon_p\big(\mathbf{w}_2^d,\kappa\big)\big] - \mathbb{E}\big[\varepsilon_p\big(\mathbf{w}_1^d,\kappa\big)\big] < \delta + \mathbb{E}\big[\varepsilon_p\big(\mathbf{z}_2^d(\alpha_2),\kappa\big)\big] - \mathbb{E}\big[\varepsilon_p\big(\mathbf{z}_1^d(\alpha_1),\kappa\big)\big] < \delta.$$

Similarly, (S7) can be concluded.

$\square$

## S6 Proofs of the Technical Lemmas

In this section, we give proofs of all the unproved lemmas stated in the paper.

### S6.1 Proof of Lemma S1

*Proof.* Inequality (S3) can be concluded from part i. of Lemma S3, stated in Section S5.5, by letting $v_l = \varepsilon_l\|\mathbf{w}_l\|$. Inequality (S4) can be concluded from (S3) and since when $\|\mathbf{w}\| \le R$, then

$$\prod_{l=1}^{L} \|\mathbf{w}_l\| \le \left(\frac{R}{\sqrt{L}}\right)^L.$$

$\square$

### S6.2 Proof of Lemma S3

*Proof.* i. Similar to [NBS18], we will show the first inequality by induction. Let $f_{\mathbf{w}}^l(\mathbf{x})$ denote the output of the $l$th layer: $f_{\mathbf{w}}^1(\mathbf{x}) = \mathbf{W}_1\mathbf{x}$ and $f_{\mathbf{w}}^l(\mathbf{x}) = \mathbf{W}_l\phi\big(f_{\mathbf{w}}^{l-1}(\mathbf{x})\big)$. We show that for $i \in [\![1, L]\!]$, following relations hold:

$$\|f_{\mathbf{w}}^i(\mathbf{x}) - f_{\mathbf{w}'}^i(\mathbf{x})\| \le B\prod_{l=1}^{i}(\|\mathbf{w}_l\| + v_l) - B\prod_{l=1}^{i}\|\mathbf{w}_l\|.$$

The induction base $i=0$ holds trivially. Assume that it holds till layer $i$. We show that it holds for layer $i+1$ as well. Note that with our notations $\mathbf{w}_l = \mathbf{vec}(\mathbf{W}_l)$ and consequently $\|\mathbf{W}_l\| = \|\mathbf{w}_l\|$.

$$
\begin{aligned}
\big\|f_{\mathbf{w}'}^{i+1}&(\mathbf{x}) - f_{\mathbf{w}}^{i+1}(\mathbf{x})\big\| \\
&= \big\|\mathbf{W}_{i+1}'\phi\big(f_{\mathbf{w}'}^i(\mathbf{x})\big) - \mathbf{W}_{i+1}\phi\big(f_{\mathbf{w}}^i(\mathbf{x})\big)\big\| \\
&= \big\|(\mathbf{W}_{i+1} + \mathbf{W}_{i+1}' - \mathbf{W}_{i+1})\big(\phi\big(f_{\mathbf{w}}^i(\mathbf{x})\big) + \phi\big(f_{\mathbf{w}'}^i(\mathbf{x})\big) - \phi\big(f_{\mathbf{w}}^i(\mathbf{x})\big)\big) - \mathbf{W}_{i+1}\phi\big(f_{\mathbf{w}}^i(\mathbf{x})\big)\big\| \\
&\le \big\|(\mathbf{W}_{i+1}' - \mathbf{W}_{i+1})\phi\big(f_{\mathbf{w}}^i(\mathbf{x})\big)\big\| + \big\|\mathbf{W}_{i+1}\big(\phi\big(f_{\mathbf{w}'}^i(\mathbf{x})\big) - \phi\big(f_{\mathbf{w}}^i(\mathbf{x})\big)\big)\big\| \\
&\quad + \big\|(\mathbf{W}_{i+1}' - \mathbf{W}_{i+1})\big(\phi\big(f_{\mathbf{w}'}^i(\mathbf{x})\big) - \phi\big(f_{\mathbf{w}}^i(\mathbf{x})\big)\big)\big\| \\
&\overset{(a)}{\le} v_{i+1}B\prod_{l=1}^{i}\|\mathbf{w}_l\| + (\|\mathbf{w}_{i+1}\| + v_{i+1})\left(B\prod_{l=1}^{i}(\|\mathbf{w}_l\| + v_l) - B\prod_{l=1}^{i}\|\mathbf{w}_l\|\right) \\
&= B\prod_{l=1}^{i+1}(\|\mathbf{w}_l\| + v_l) - B\prod_{l=1}^{i+1}\|\mathbf{w}_l\|.
\end{aligned}
$$

where $(a)$ is concluded since $\phi$ is 1-Lipschitz, $\phi(0) = 0$, and since due to the structure of $f_\mathbf{w}$, $\|f_\mathbf{w}^i(\mathbf{x})\|$ can be upper bounded as

$$\|f_\mathbf{w}^i(\mathbf{x})\| \leq \|\mathbf{x}\| \prod_{l=1}^{i} \|\mathbf{w}_i\| \leq B \prod_{l=1}^{i} \|\mathbf{w}_i\|.$$

Next, we show the second inequality.

$$
\begin{aligned}
\left|\ell_{0,\tau}(\mathbf{z}, f_\mathbf{w}) - \ell_{0,\tau}(\mathbf{z}, f_{\mathbf{w}'})\right| &\leq \frac{1}{\tau}\left|f_\mathbf{w}(\mathbf{x})[y] - \max_{j \neq y} f_\mathbf{w}(\mathbf{x})[j] - f_{\mathbf{w}'}(\mathbf{x})[y] + \max_{j' \neq y} f_{\mathbf{w}'}(\mathbf{x})[j']\right| \\
&\leq \frac{1}{\tau}|f_\mathbf{w}(\mathbf{x})[y] - f_{\mathbf{w}'}(\mathbf{x})[y]| + \frac{1}{\tau}\left|\max_{j \neq y} f_\mathbf{w}(\mathbf{x})[j] - \max_{j' \neq y} f_{\mathbf{w}'}(\mathbf{x})[j']\right| \\
&\leq \frac{1}{\tau}|f_\mathbf{w}(\mathbf{x})[y] - f_{\mathbf{w}'}(\mathbf{x})[y]| + \frac{1}{\tau}\max_{j \neq y}|f_\mathbf{w}(\mathbf{x})[j] - f_{\mathbf{w}'}(\mathbf{x})[j]| \\
&\overset{(a)}{\leq} \frac{\sqrt{2}}{\tau}\|f_\mathbf{w}(\mathbf{x}) - f_{\mathbf{w}'}(\mathbf{x})\|,
\end{aligned}
$$

where $(a)$ is derived using the relation $x + y \leq \sqrt{2(x^2 + y^2)}$, for $x, y \in \mathbb{R}_+$.

ii. To show the first inequality, note that due to symmetry, R.H.S. of part i. is maximized when $\|\mathbf{w}_l\| = R/\sqrt{L}$, for $l \in [\![1, L]\!]$. Hence,

$$\|f_\mathbf{w}(\mathbf{x}) - f_{\mathbf{w}'}(\mathbf{x})\| \leq B\left(\left(\frac{R}{\sqrt{L}} + \upsilon\right)^L - \left(\frac{R}{\sqrt{L}}\right)^L\right). \tag{S44}$$

Next, we show that if $a \geq b \geq 0$ and $n \in \mathbb{N}$, then

$$a^n - b^n \leq n(a - b)a^{n-1}. \tag{S45}$$

We show this by induction. It trivially holds for $n = 1$. Suppose that it holds till $n \leq i - 1$. We show that it holds for $n = i$, as well.

 – if $i$ is even, then

$$a^i - b^i = \left(a^{\frac{i}{2}} - b^{\frac{i}{2}}\right)\left(a^{\frac{i}{2}} + b^{\frac{i}{2}}\right) \overset{(a)}{\leq} \frac{i}{2}(a - b)a^{i/2-1} \times 2a^{i/2} = i(a - b)a^{i-1},$$

 where $(a)$ is derived using the induction assumption.
 – if $i$ is odd, then

$$a^i - b^i = (a - b)\sum_{k=0}^{i-1} a^k b^{i-1-k} \leq i(a - b)a^{i-1}.$$

Thus, using (S44) and (S45) and since $\upsilon \geq R/\sqrt{L}$,

$$\|f_\mathbf{w}(\mathbf{x}) - f_{\mathbf{w}'}(\mathbf{x})\| \leq BL\left(\frac{2R}{\sqrt{L}}\right)^{L-1}\upsilon.$$

This completes the proof for the first inequality. Finally, the second inequality trivially follows from the first one and part i.

$\square$

### S6.3 Proof of Lemma S4

*Proof.* Note that there exists $\binom{d}{\kappa d}$ different ways to choose $\kappa d$ coordinates with zero values. Next, each of the resulting $\kappa d$-dimensional sub-space can be discretized using at most $\left(\frac{3R(\delta)}{\upsilon}\right)^{\kappa d}$ number of points due to [Wu20, Theorem 14.2.]. Using the following lemma completes the proof.

**Lemma S7** ([Gal68, Exercise 5.8.b.]). *For $n, m \in \mathbb{Z}^+$ and $m \leq n$, $\binom{n}{m} \leq e^{nh_b(m/n)}$.*

$\square$

## S6.4   Proof of Lemma S5

*Proof.* First we show that in general when random variables $y_i$, $i \in [\![1, m]\!]$ are independent and $\sum_{i=1}^{m} a_i \leq a$, then

$$\mathbb{P}\left(\sum_{i=1}^{m} y_i \geq a\right) \leq \sum_{i=1}^{m} \mathbb{P}(y_i \geq a_i).$$

We prove this for the case of $m = 2$, and the general case follows by an induction.

$$\begin{aligned}
\mathbb{P}(y_1 + y_2 \geq a) &= \mathbb{P}(y_1 + y_2 \geq a, y_1 \geq a_1) + \mathbb{P}(y_1 + y_2 \geq a, y_1 < a_1) \\
&\leq \mathbb{P}(y_1 \geq a_1) + \mathbb{P}(y_1 + y_2 \geq a, y_1 < a_1) \\
&\leq \mathbb{P}(y_1 \geq a_1) + \mathbb{P}(y_2 \geq a - a_1, y_1 < a_1) \\
&\leq \mathbb{P}(y_1 \geq a_1) + \mathbb{P}(y_2 \geq a_2).
\end{aligned}$$

Next, since stable distributions are continuous distributions, hence $P(\|\mathbf{x}\| \geq R(\delta)) = \delta$.

Now, to show the idea, first show that if $\mathbf{x}$ is an i.i.d. $d$-dimensional vector with $\mathcal{S}\alpha\mathcal{S}_n(\sigma)$ distributions and $\alpha \in (1, 2)$, then for $\delta < 2d(2 - \alpha)^{\alpha}$, $R(\delta)$ can be bounded as

$$R(\delta) \leq 3\sigma\sqrt{d}\left(\frac{4d}{\delta}\right)^{1/\alpha}. \tag{S46}$$

To show this,

$$\begin{aligned}
\delta = P\left(\|\mathbf{x}\|^2 \geq R^2(\delta)\right) &\leq \sum_{i=1}^{d} P\left(\|\mathbf{x}_i\|^2 \geq R^2(\delta)/d\right) \\
&= \sum_{i=1}^{d} P\left(\|\mathbf{x}_i\| \geq R(\delta)/\sqrt{d}\right) \overset{(a)}{\leq} 4d\left(\frac{3\sigma\sqrt{d}}{R(\delta)}\right)^{\alpha},
\end{aligned} \tag{S47}$$

where $(a)$ holds when $R(\delta) \geq 4\sigma\sqrt{d}/(2 - \alpha)$ due to the following inequality from [BŁM20, Theorem 19]. The result is stated for a $\mathcal{S}\alpha\mathcal{S}(\sigma_\alpha) \equiv \mathcal{S}\alpha\mathcal{S}_n(1)$ distribution, where $\sigma_\alpha := (2\Gamma(-\alpha)\cos((2 - \alpha)\pi/2))^{1/\alpha}$. Here, we state the result for arbitrary $\mathcal{S}\alpha\mathcal{S}_n(\sigma)$. If $y \sim \mathcal{S}\alpha\mathcal{S}_n(\sigma)$ and $\alpha \in (1, 2)$, then for $a \geq 4\sigma/(2 - \alpha)$

$$\mathbb{P}(y \geq a) \leq \frac{16}{3}\left(\frac{2\sigma}{a}\right)^{\alpha} \leq 4\left(\frac{3\sigma}{a}\right)^{\alpha}.$$

Re-arranging (S47) and considering the condition $R(\delta) \geq 4\sigma\sqrt{d}/(2 - \alpha)$, yields

$$R(\delta) \leq \max\left(3\sigma\sqrt{d}\left(\frac{4d}{\delta}\right)^{1/\alpha}, \frac{4\sigma\sqrt{d}}{(2 - \alpha)}\right).$$

Hence, (S46) holds, at least when

$$3\sigma\sqrt{d}\left(\frac{4d}{\delta}\right)^{1/\alpha} \geq \frac{4\sigma\sqrt{d}}{(2 - \alpha)},$$

which is satisfied when $\delta < 2d(2 - \alpha)^{\alpha}$.

Now, to show the lemma, let $a_l := d\sigma^2/\sigma_l^2$. Then, similar steps concludes

$$\delta \leq \sum_{l=1}^{L}\sum_{i=1}^{d_l} \mathbb{P}\left(\|x_{l,i}\| \geq \frac{R(\delta)}{\sqrt{a_l}}\right) \overset{(a)}{\leq} 4\sum_{l=1}^{L}\sum_{i=1}^{d_l}\left(\frac{3\sigma\sqrt{d}}{R(\delta)}\right)^{\alpha_l} \overset{(b)}{\leq} 4d\left(\frac{3\sigma\sqrt{d}}{R(\delta)}\right)^{\alpha}, \tag{S48}$$

where $(a)$ holds when $R(\delta) \geq \max_l 4\sigma\sqrt{d}/(2 - \alpha_l)$ and $(b)$ holds when $R(\delta) \geq 3\sigma\sqrt{d}$. Note that $3\sigma\sqrt{d} \leq 4\sigma\sqrt{d}/(2 - \alpha_l)$. Finally, similarly, (S46) holds if $\delta < 2d(2 - \max_l \alpha_l)^{\alpha}$.

$\square$

## S6.5  Proof of Lemma S6

*Proof.* To upper bound $\mathcal{N}'_v$, first consider the space $\mathcal{W}''$, defined as[5]

$$\mathcal{W}'' := \bigcup_{\substack{\mathcal{A} \\ |\mathcal{A}|=\kappa d}} \mathcal{W}''_{\mathcal{A}},$$

$$\mathcal{W}''_{\mathcal{A}} := \left\{ \mathbf{w} \in \mathbb{R}^d \,\big|\, \|\mathbf{w}_{\mathcal{A}}\| \le R(\delta), \|\mathbf{w}_{\mathcal{A}^C}\| \le \varepsilon R(\delta) \right\}.$$

Since each of $\mathcal{W}''_{\mathcal{A}}$ is a convex space, then if $v < \varepsilon R(\delta)$, by [Wu20, Theorem 14.2.], it can be discretized with $v$-precision using at most

$$\left(\frac{3}{v}\right)^d \frac{\mathrm{Vol}(\mathcal{W}''_{\mathcal{A}})}{\mathrm{Vol}(\mathbb{B}_d)} = \left(\frac{3}{v}\right)^d \frac{\frac{\pi^{\kappa d/2} R^{\kappa d}}{\Gamma(\kappa d/2+1)} \times \frac{\pi^{(1-\kappa)d/2}(\varepsilon R)^{(1-\kappa)d}}{\Gamma((1-\kappa)d/2+1)}}{\frac{\pi^{d/2} R^d}{\Gamma(d/2+1)}}$$

$$= \left(\frac{3\varepsilon^{(1-\kappa)}R}{v}\right)^d \frac{\Gamma(d/2+1)}{\Gamma(\kappa d/2+1)\Gamma((1-\kappa)d/2+1)},$$

number of points, where $\mathbb{B}_d$ is the $d$-dimensional unit ball. Now, since $\mathcal{W}(R(\delta), \varepsilon, d, \kappa) \subseteq \mathcal{W}''$,

$$\mathcal{N}'_v \le \binom{d}{\kappa d} \left(\frac{3\varepsilon^{(1-\kappa)}R}{v}\right)^d \frac{\Gamma(d/2+1)}{\Gamma(\kappa d/2+1)\Gamma((1-\kappa)d/2+1)},$$

$$\overset{(a)}{\le} e^{d\left(h_b(\kappa)+h_b^{(1)}(\kappa)\right)} \left(\frac{3\varepsilon^{(1-\kappa)}R}{v}\right)^d$$

$$\overset{(b)}{=} \left(\frac{3\rho_\varepsilon(\kappa,d)R}{v}\right)^d, \tag{S49}$$

where $(a)$ is concluded from Lemma S7 and the following lemma and $(b)$ is concluded since one way to discretize $\mathcal{W}(R(\delta), \varepsilon, d, \kappa)$ is to consider the whole sphere with radius $R$, which needs at most $(3R/v)^d$, due to [Wu20, Theorem 14.2.].

**Lemma S8.** *For $n, m \in \mathbb{Z}^+$, $n \ge m$, and $n \ge 10$,*

$$\frac{\Gamma(n/2+1)}{\Gamma(m/2+1)\Gamma((n-m)/2+1)} \le e^{\lceil n/2 \rceil \max(h_b(\lceil m/2 \rceil/\lceil n/2 \rceil), h_b(\lfloor m/2 \rfloor/\lceil n/2 \rceil))}.$$

$\square$

## S6.6  Proof of Lemma S8

*Proof.* For $m = 0$ or $m = n$, the claim holds with equality. Let $1 \le m \le n - 1$. When $n$ and $m$ are even, then the lemma can be concluded from Lemma S7. Assume, at least one of $n$ and $m$ are odd numbers. We consider two cases of $n$ being odd and even separately.

Note that for $a \in \mathbb{N}$, [Rob55]

$$\Gamma(a+1) = \sqrt{2\pi a}\, a^a e^{-a} e^{r_a}, \tag{S50}$$

where $1/(12a+1) < r_a < 1/(12a)$. Moreover,

$$\Gamma\left(a+\frac{1}{2}\right) = \frac{\sqrt{\pi}(2a)!}{4^a a!} \overset{(*)}{=} \sqrt{2\pi}\, a^a e^{-a} e^{s_a}, \tag{S51}$$

where $(*)$ is derived using (S50) with $s_a$ being bounded as

$$\frac{1}{24a+1} - \frac{1}{12a} < s_a < \frac{1}{24a} - \frac{1}{12a+1}.$$

---

[5]For a set $\mathcal{A} = \{i_1, \ldots, i_r\} \subseteq [\![1, d]\!]$, denote $\mathbf{x}_{\mathcal{A}} := (x_{i_1}, x_{i_2}, \ldots, x_{i_r})$.

Odd $n$: Let $n = 2k + 1$ and $m = 2q$, where $1 < q \leq k$. Then,

$$\frac{\Gamma(n/2+1)}{\Gamma(m/2+1)\Gamma((n-m)/2+1)} \overset{(a)}{=} \frac{e^{s_{k+1}-(r_q+s_{k+1-q})}(k+1)^{k+1}}{\sqrt{2\pi q}\, q^q(k+1-q)^{k+1-q}}$$

$$\overset{(b)}{\leq} \frac{(k+1)^{k+1}}{q^q(k+1-q)^{k+1-q}}$$

$$< e^{(k+1)h_b(q/(k+1))}$$

$$< e^{\lceil n/2 \rceil \max(h_b(m/2\lceil n/2\rceil), h_b(m/2\lceil n/2\rceil))}, \qquad \text{(S52)}$$

where $(a)$ is derived using (S50) and (S51), and $(b)$ is derived, since

$$s_{k+1}-(r_q+s_{k+1-q}) < \frac{1}{24k+24} - \frac{1}{12k+13} - \frac{1}{12q+1} - \frac{1}{24(k-q)+25} + \frac{1}{12(k-q)+12}$$

$$\leq \frac{1}{24k+24} - \frac{1}{12k+13} - \frac{1}{12k+1} - \frac{1}{25} + \frac{1}{12}$$

$$< \frac{1}{12} - \frac{1}{25} \leq 0.05.$$

The case of $m$ being odd is similar.

Even $n$: Let $n = 2k$ and $m = 2q + 1$, where $1 < q < k$ and $k \geq 5$. Then,

$$\frac{\Gamma(n/2+1)}{\Gamma(m/2+1)\Gamma((n-m)/2+1)} \overset{(a)}{=} \frac{e^{1+r_k-(s_{q+1}+s_{k-q})}\sqrt{k}\, k^k}{\sqrt{2\pi}(q+1)^{q+1}(k-q)^{k-q}}$$

$$\overset{(b)}{<} \frac{1.17\sqrt{k}\, k^k}{(q+1)^{q+1}(k-q)^{k-q}} \qquad \text{(S53)}$$

where $(a)$ is derived using (S50) and (S51) and $(b)$ is derived since

$$r_k-(s_{q+1}+s_{k-q}) < \frac{1}{12k} - \frac{1}{24q+25} + \frac{1}{12q+12} - \frac{1}{24(k-q)+1} + \frac{1}{12(k-q)}$$

$$\leq \frac{1}{6k} - \frac{1}{24k+1} - \frac{1}{25} + \frac{1}{12}$$

$$< 0.07,$$

where the last step holds for $k \geq 5$.

Next, for $k \geq 5$, either $q+1 \geq 1.17\sqrt{k}$ or $k-q \geq 1.17\sqrt{k}$. Otherwise, we would conclude $k+1 < 2.34\sqrt{k}$, which is a contradiction for $k \geq 5$.

- If $q+1 \geq 1.17\sqrt{k}$, then (S53) is upper bounded by

$$\frac{1.17\sqrt{k}\, k^k}{(q+1)^{q+1}(k-q)^{k-q}} \leq \frac{k^k}{(q+1)^{q}(k-q)^{k-q}}$$

$$\leq \frac{k^k}{q^q(k-q)^{k-q}}$$

$$\leq e^{kh_b(q/k)} = e^{\frac{n}{2}h_b(2\lfloor m/2\rfloor/n)}.$$

- If $k-q \geq 1.17\sqrt{k}$, then (S53) is upper bounded by

$$\frac{1.17\sqrt{k}\, k^k}{(q+1)^{q+1}(k-q)^{k-q}} \leq \frac{k^k}{(q+1)^{q+1}(k-q)^{k-q-1}}$$

$$\leq \frac{k^k}{(q+1)^{q+1}(k-q-1)^{k-q-1}}$$

$$\leq e^{kh_b((q+1)/k)} = e^{\frac{n}{2}h_b(2\lceil m/2\rceil/n)}.$$

$\square$