# OpenReview forum: "Heavy Tails in SGD and Compressibility of Overparametrized Neural Networks"
_NeurIPS.cc/2021/Conference — NeurIPS 2021 Poster_

### Official Review · Reviewer_9D1o · 2021-07-13

**Rating:** 6
**Confidence:** 3

**Summary:**

The paper looks at the compressibility and generalization ability of multi-layer neural networks under the heavy tailed mean-field limit. The mean-field condition is invoked to simplify the analysis of the distribution of weight parameters as the distributions of individual weights are independent, while the heavy-tailed condition is needed to ensure an abundance of small weights (weights at the tail) whose removal would result in bounded relative error. Compression error is defined in weight space as the relative distance between the compressed weight vector (weight vector where small weights are set to zero) and the original weight vector.

The paper shows that arbitrarily low compression errors can be achieved under a range of pruning strategies (global pruning, layer-wise pruning, singular value pruning, and neuron pruning) if the network is sufficiently large. Another condition is for the p-norm used to calculate the error to have p larger than the heavy tail coefficients.


**Limitations And Societal Impact:**

The paper has no negative societal effects that I am aware of.

**Main Review:**

The paper is well-written and the authors are to be commended for providing the necessary background for making their theoretical contributions accessible to readers who are new to this subject area. I believe the theoretical results are novel, even if the main results follow in a rather straightforward manner from a standard result on the compressibility of heavy-tailed i.i.d random variables (Lemma S2 in the appendix). In general the compressibility results are largely intuitive: The larger the number of weights is, and the more heavy-tailed their distribution is, the more probable that there will be enough small weights so that their removal would have minimal impact on the weight vector norm. The results are thus not surprising, especially that they follow from standard results in the statistics of heavy-tailed distributions.

In the empirical results, the authors convincingly demonstrate that in several cases, there is a correlation between the tail indices and the prunability and generalization ability of the network.
However, the authors do not control the tail indices directly. Rather, they vary the learning rate and batch size to obtain the desired tail indices. As such, the empirical results do not categorically prove that tail indices have a causal influence on the prunability and generalization ability of the network. There could be other confounding factors here as the tail indices are not controlled directly. The authors, however, phrase the discussion as if heavier tail indices lead to better generalization and prunability. The empirical evidence certainly points in that direction, but it is not categorical. The authors need to either 1)devise experiments where they can directly control tail indices while other factors are constant and observe the effects, 2) modify the discussion to highlight that the empirical results are circumstantial (but not definite) evidence that heavier tails are the causal factors leading to more prunable and generalizable networks in the experimental results.

Finally, it is not clear what the practical benefits of the theoretical results are. The effect of $\eta/b$ (ratio between learning rate and batch-size) on generalization accuracy is known, as is the fact that a larger ratio leads to heavier tailed weight . While deeper theoretical understanding is a worthy goal in itself, the paper would benefit from discussing any novel practical implications of the theoretical results.



Minor comments:
1)L198: remove duplicate $\epsilon > 0$
2)L357: remove confirm


**Time Spent Reviewing:**

3 hours

---

> ### Author Response · Authors · 2021-08-09
> **Comments on Reviewer 9D1o's Feedback**
>
> We thank the reviewer for their valuable and detailed feedback.
>
> While we agree with the reviewer (and also acknowledge in the paper) that we repeatedly use a seminal result from compressive sensing to develop our theory, we believe that making the connection between hyperparameters of SGD ($\eta, b$) and the prunability of the network, using $\ell_p$ compressibility results, was quite unexpected: _if one uses different step-size or batch-size, the SGD algorithm might provide outputs with different compressibility properties_.
>
> **Regarding the causal interpretation of the experimental results:** We thank the reviewer for bringing up this fruitful point of discussion. [GSZ20, Theorem 4, Proposition 5] shows that the tail index is fully determined by $\eta$, $b$, and the Hessian of the problem in the context of quadratic optimization: the tail-index changes *monotonically* with respect to both $\eta$ and $b$. We establish the relationship empirically in our paper in the context of neural networks, replicating the results presented in [GSZ20]. In our paper, we also show that the existence of heavy-tailed network parameters leads to compressibility [Theorems 1-3], and thus to arbitrarily small perturbation in the network outputs when pruned [Lemma S1]. We also demonstrate that the more compressible the network is, the smaller its generalization bound is [Theorem 4].
>
> Using a different, geometric framework, [SSDE20, Appendix S1.2] (arxiv:2006.09313) experimentally demonstrated that a lower tail-index leads to a better generalization, where they directly varied the tail-index and monitored the generalization error, as the reviewer requested. Given these results and our experimental findings, we believe that a causal interpretation of the relationships in question is not without support.
>
> As the reviewer notes, experimental manipulation of tail index directly in the case of neural networks trained with real data is hard to formulate and conduct. However, to make our argument stronger, in the supplement of the final version of the paper, we will include synthetic experiments that extend the ones of [SSDE20], to demonstrate the relationship between the tail index, prunability, and generalization error by generating network parameters with given tail-indices, as well as noisy observations to show how prunability changes with decreasing tail index. We will also provide a more detailed discussion of the points made above.
>
> **Regarding our work’s practical implications:** Although the relationship between $\eta/b$ ratios and generalization behavior is examined in different mathematical frameworks, its relation to generalization through compressibility has not been investigated previously to the best of our knowledge. In our opinion, that changing $\eta/b$ leads to a more compressible network is an unexpected finding with practical implications as well as theoretical value. One immediate practical implication would be, for example, if a practitioner trains a model for deployment in a resource-constrained environment (e.g., mobile phone), they would benefit from training the network with higher $\eta/b$, up to the limit that the training converges. This would lead to not only better-expected generalization performance, but also to higher prunability, and therefore would result in increased space- and time-efficiency without considerable performance loss. We will include a short discussion to illustrate this point.
>
> **Regarding the minor comments:**
> 1. We could not unfortunately find the duplicate. The reviewer might be referring to $\varepsilon$ and $\epsilon$. The former one is used for compression loss ($\lVert\mathbf{w}^{\kappa d} - \mathbf{w}\rVert_p \leq \varepsilon \lVert\mathbf{w}\rVert_p$) and the latter for the probability margin ($1-\epsilon$).
> 2. Thank you, we will make the necessary corrections.

---

### Official Review · Reviewer_jjg2 · 2021-07-16

**Rating:** 6
**Confidence:** 3

**Summary:**

The main contribution of this paper is to link the two properties of compressibility and SGD. (i) as the network size goes to infinity, the system can converge to a mean-field limit, where the network weights behave independently, (ii) for a large step-size/batch-size ratio, the SGD iterates can converge to a heavy-tailed stationary distribution. In the case where these two phenomena occur simultaneously, we prove that the networks are guaranteed to be ‘lp-compressible’. Further experiments on fully connected and convolutional neural networks and show that the results are in strong accordance with the theory.

**Limitations And Societal Impact:**

1.	The theoretical framework proposed in this article is not applied to the Resnet layer, which is widely used in recent works.
2.	This paper only experimented on the mnist and cifar10 datasets, which seems relatively small for pruning experiments.

**Main Review:**

Originality: Novel. This work links compressibility with SGD, and theoretically explains that simple pruning strategies in some networks are also effective.

Quality: The theoretical proof is quite sufficient.

Clarity: well written.

Significance: Moderately significant. This paper does theoretical derivation and experimental verification on the convolutional layer and the fully connected layer.

**Time Spent Reviewing:**

10 hours

---

> ### Author Response · Authors · 2021-08-09
> **Comments on Reviewer jjg2's Feedback**
>
> We thank the reviewer for the positive and beneficial feedback.
>
> **Regarding the application to the Resnet layer and choices of dataset:** We agree with the reviewer that testing our hypotheses on other neural network layers and/or larger datasets would be an important addition. However, due to the theoretical work and detailed experimentation our hypotheses required, we had to limit our work to the settings presented in the paper.
>
> More specifically, each model/dataset combination requires establishing the following relationships experimentally:
> 1. the relationship between $\eta/b$ and estimated tail index,
> 2. the relationship between the tail index and prunability for different methods of pruning,
> 3. the relationship between the tail index and generalization.
>
> Therefore the number of experiments increases combinatorially with each model or dataset introduced. In the final version of the paper, we will also include experiments with CIFAR100 and include these in the supplementary material to provide more empirical demonstration regarding our hypotheses. Extending our theoretical and empirical analyses to different types of layers and model architectures is an exciting direction for future research and we will provide a more detailed discussion of these in the final section of our paper.

---

### Official Review · Reviewer_uErr · 2021-07-19

**Rating:** 7
**Confidence:** 3

**Summary:**

This paper analyze the compressibility of over-parameterized neural networks using the heavy-tail property of SGD. The authors show that when the network weights are approximately sampled from a heavy-tailed distribution, the weights are more compressible. The authors also provide a generalization guarantee showing that more compressible networks have smaller generalization gap.

**Limitations And Societal Impact:**

This paper mentioned a few possible future directions which can be considered as resolving the limitations of the current work.
I did not observe immediate societal impact.

**Main Review:**

I think overall this is a good paper. I like the connection between the heavy-tail property of SGD on large network and the compressibility of the network (the l_p compressibility from the compressed sensing literature). The results seem to be new in the literature and I believe the results will strengthen our understanding of neural network compression.

The paper is clearly written. The notation is slightly heavy (which happens to most neural network theory papers) but mostly clearly defined. I am able to follow most parts of the paper.

I have some concerns on the generalization bound in Theorem 4. It seems to say that if the network has smaller kappa (more compressible), then its generalization gap will be smaller. However, this is only an upper bound for the population risk of the compressed network (R_0(\hat{w})). When kappa decreases, the upper bound decreases, but it does not necessarily mean R_0(\hat{w}) also decreases. The authors should either prove a lower bound or clearly state this limitation in the paper.

Minor:

Line 85: Y is defined as a discrete set of integers {1, …, d_Y} on line 81. So f_w(x) should be a function that maps from X_B to R^{d_Y} rather than Y; similar thing on line 93.

Line 212-213: “how would the smallest kappa_l would differ…” it seems to be a grammar error—two “would”.


**Time Spent Reviewing:**

2

---

> ### Author Response · Authors · 2021-08-09
> **Comments on Reviewer uErr's Feedback**
>
> We thank the reviewer for their constructive comments.
>
> **Regarding the generalization bound in Theorem 4:** We do agree with the reviewer. We will modify the parts that could lead to this misunderstanding and highlight this limitation, which exists also in previous works, for example in [AGNZ18, SAM+20, SAN20, HJTW21]. Unfortunately, proving a lower bound is a notoriously difficult problem. Instead, we illustrated this relation between compressibility and generalization performance experimentally in Section 5.
>
> **Regarding the minor comments:** Thank you for the detailed reading of the paper. We will make the necessary modifications.

---

> > ### Comment · Reviewer_uErr · 2021-08-29
> > **Thanks for your response**
> >
> > Thanks for your response. I think clarifying the generalization bound is helpful. I'll keep my score.

---

### Official Review · Reviewer_9G5Z · 2021-07-20

**Rating:** 6
**Confidence:** 4

**Summary:**

This paper gives a theoretical characterization of the underlying cause that makes the neural networks amenable to common compression schemes in the over-parameterized regime. It is done by connecting the dynamics of the learning algorithms such as SGD to the heavy tails of the weights of a network, which leads to better compressibility. The main contribution is to provide sufficient conditions for the compressibility from the limiting distribution of SGD, and show how the compressibility of a network can lead to better generalization.


**Ethics Review Area:**

["I don’t know"]

**Limitations And Societal Impact:**

yes

**Main Review:**

The paper is very well written, and is addressing an important topic of the field of why and when a pruning method is useful. It also makes several technical contributions by borrowing classical results to prove new theorem. However, the following points need to be considered:

My main concern is that the main result is based on a condition / conjecture (Definition 1) that seems to be too ideal. The remark on l.163  is not exact: note that [DBDFS20] proved that a similar form of (4)  indeed holds … I think that it remains still an open question. In DBDFS20, L=2, and W_L is fixed, only W_1 is considered to be learnt by SGD. Thus it is very hard to believe why (4) holds in more complicated cases such as the heavy tails. Such kind of details needs to be mentioned so that your readers have a clear view of the current state-of-the-art.

About the generalization bounds, what are the contributions compared to the work mentioned on line 275? Are you using a very different technique to prove the similar results? How similar the Theorem 4 compared to the existing results?

The assumption on SGD in Definition 1 is for eq. (2), however, in the reference GSZ20, it seems that ergodic averaging SGD is proved to converge to a stable distribution (l.340). Is the SGD limit of (2) also converging to a heavy-tailed distribution? Why in the numerical results, the ergodic averaging SGD is chosen rather than (2) for evaluation?

Theorem 2 eq. (5), the w_l should be W_l as the norm ||. || is for Frobenius norm, thus there is a typo? It would be good to give some intuition why there is division by 2 in the condition p>=max_l alpha_l / 2. This is not the case for Theorem 1 and 3.

Which p norm is taken in Figure 3 to compute the compression error?

line 323, what it means to be ‘consistent’ with the results of … ?

In Figure 1-7, sometimes write Mean estimated tail index, sometime write estimated tail index, are they the same?



**Time Spent Reviewing:**

6

---

> ### Author Response · Authors · 2021-08-09
> **Comments on Reviewer 9G5Z's Feedback**
>
> We thank the reviewer for the insightful comments and discussion.
>
> **Regarding Definition 1:** We agree with the reviewer that proving the required property in Definition 1 is not straightforward and as the reviewer mentioned in [DBDFS20] the mean-field property is proven for a two-layer network with fixed weights for one of the layers. We will detail this point and make the state-of-the-art more clear. Nevertheless, we shall reiterate that the form in Definition 1 is not a necessary condition and can be relaxed in different ways, as we summarized in Section 3.3, where the relaxed versions might require simpler analysis. On the other hand, we shall note that heavy-tails do not prevent the emergence of propagation of chaos, as illustrated in [JMW08] and [LMW19], and we believe that by using the tools developed in [GSZ20, HM20] and combining it with the setup presented in [DBDFS20] (two-layered network where one of the layers is fixed), we might be able to show that a version of Definition 1 holds. Yet, we agree with the reviewer that this is an open problem and we will expand our discussion in L163 accordingly.
>
>
> **Regarding the generalization bound:** The similarities between our results and the mentioned works are mainly in using the “compression” approach (assuming that the network is “compressible”) to establish a generalization bound (either on the original network or the compressed network) and also by considering the empirical loss with margin. However, the results established in [AGNZ18, SAM+20, SAN20, HJTW21] are obtained by either making specific compressibility assumptions or considering specific pruning strategies that are not compatible with our theoretical pruning framework which is based on $\ell_p$-compressibility. In particular, in our generalization bound one can observe that using, for example, layer-wise magnitude pruning strategy, the compression ratio is manifested as the factor $\approx\sqrt{\kappa}$ in the generalization bound, showing that a more compressed network has a better generalization bound. Such a factor cannot be observed in other mentioned works, as they consider compressibility approaches that are different from $\ell_p$-compressibility. As an example, in [Theorem 2.1, AGNZ18], a generalization bound is established on the compressed network by assuming that it has a finite hypothesis class. This cannot be applied in our work, as the compressed network using $\ell_p$-compressibility does not have a finite hypothesis class. We will add a discussion to address these connections.
>
> **Regarding the limit distribution of SGD:** The reviewer is right, the limiting distribution of SGD (2) also converges to a heavy-tailed distribution, but not necessarily an alpha-stable distribution. The convergence of SGD to a heavy-tailed stationary distribution is proven in both [HM20, Theorem 1] and [GSZ20, Theorem 1]. In addition, in [GSZ20, Corollary 9], as the reviewer mentioned, it is shown that a properly scaled sum of weights over iterations converges to an $\alpha$-stable distribution.
>
> Estimating the tail-index of an alpha-stable variable is arguably easier than estimating the tail-index of a general heavy-tailed random variable. Hence, in the experiments, we use the ergodic averaging (which does not change the tail index of the parameters) only for estimating the tail index of the resulting parameters to conform with the assumptions of our estimator. The rest of the experiments have been conducted without ergodic averaging. We also observe that the results with/without ergodic averaging are virtually identical in both tail index estimation tasks and pruning experiments; we will include these results in the supplementary material. We will make these points clearer in the final version of the text.
>
> **Regarding the notation in Theorem 2:** In the notations section, we have stated that for a vector $\mathbf{x}$, $\lVert\mathbf{x}\rVert$ implies $\lVert\mathbf{x}\rVert_2$. Hence, the used notation in the theorem is correct. Indeed, this could be alternatively stated by using matrices $\mathbf{W}_l$ and the norm $\lVert \cdot \rVert$, which would imply the Frobenius norm in this case. However, since all compression results are stated for the vectorized version of the weights, here also we have stated it for $\mathbf{w}_l$.
>
> **Regarding $p \geq \max_l \alpha_l/2$ in Theorem 2:** The reason is that Theorem 2 is based on pruning the eigenvalues of matrices $h_l^{-2 / \alpha} \mathbf{W}\_l^T \mathbf{W}\_l$ and moreover in general if a matrix $\mathbf{A} \in \mathbb{R}^{n \times m}$ has elements $[\mathbf{A}]_{i,j}$ that are independent and identically distributed from the symmetric $\alpha$-stable distribution, then by [TTR+20, Theorem 2.7], as $m \rightarrow \infty$, the eigenvalues of $m^{-2 / \alpha} \mathbf{A}^T \mathbf{A}$ weakly converge to independent random variables, that are identically distributed from a positive stable distribution with tail-index $\alpha/2$. This is the reason why the variable $p$ needs to be larger than $\alpha/2$, rather than $\alpha$. We will add a short discussion for this.
>
> **Regarding Figure 3:** In all our experiments we use $p=2$; we will make this clear at the beginning of the experiments section.
>
> **Regarding line 323:** In [HJTW21, KLG+21] it is shown that if a neural network is “compressible”, then this network has a good generalization performance (bound), but their framework does not handle $\ell_p$ -compressibility. In Theorem S6 in the supplementary document, we derived a generalization bound for the original network, which is compatible with their findings: if a network is more compressible in the sense of $\ell_p$-compressibility, not only the generalization bound for the compressed network but also for the original network improves. This connection is highlighted in Section S2 of the supplement.
>
> **Regarding mean estimated tail index vs estimated tail index:** In all figures except Figure 3, we refer to the mean estimated tail index. In Figure 3, on the other hand, we refer to the estimated tail index of each layer as the figure includes all layers from all trained models. In the final version of the paper, we will make the captions of Figures 4, 5, and 6 clearer to resolve this issue.

---

### Author Response · Authors · 2021-08-09
**Thank you to all the reviewers**

We thank the reviewers very much for their time and valuable feedback. We are excited to see that all the reviewers have positive opinions regarding our work and find it interesting and well-presented. We address the reviewers' feedback as separate comments below.

---

### Decision · Program_Chairs · 2021-09-27

**Decision:**

Accept (Poster)

**Comment:**

Four reviewers recommend this paper to be accepted. The paper provides result addressing why and when pruning is possible. A concern is that the main result is based on a condition that might be too ideal. Taking the discussion into consideration, I find that the manuscript makes valuable contributions outweighing its limitations. Hence I am recommending the submission for publication. I ask the authors to implement the improvements promised in their responses and to carefully consider the reviewers comments in the preparation of the final manuscript.